Subject Areas:
mathematical modelling/complexity/ biogeography

Keywords:
economic complexity, collective know-how, industry complexity, urban employment

Author for correspondence:
Andres Gomez-Lievano
e-mail: andres_gomez@hks.harvard.edu

# Estimating the drivers of urban economic complexity and their connection to economic performance

Andres Gomez-Lievano[1,2] and Oscar Patterson-Lomba[2]

[1]Growth Lab, Harvard University, Cambridge MA, USA
[2]Analysis Group Inc., Boston MA, USA

(iD) AG-L, 0000-0001-8320-0857

Estimating the capabilities, or inputs of production, that drive and constrain the economic development of urban areas has remained a challenging goal. We posit that capabilities are instantiated in the complexity and sophistication of urban activities, the know-how of individual workers, and the city-wide collective know-how. We derive a model that indicates how the value of these three quantities can be inferred from the probability that an individual in a city is employed in a given urban activity. We illustrate how to estimate empirically these variables using data on employment across industries and metropolitan statistical areas in the USA. We then show how the functional form of the probability function derived from our theory is statistically superior when compared with competing alternative models, and that it explains well-known results in the urban scaling and economic complexity literature. Finally, we show how the quantities are associated with metrics of economic performance, suggesting our theory can provide testable implications for why some cities are more prosperous than others.

## 1. Introduction

Fine-grained representations of the economy of countries, regions and cities in terms of what they produce, and the industries they have, have revealed that the location of economic activities across places is not random [1–6]. Instead, activities tend to locate in cities of different sizes depending on the number of inputs they require [7,8] and co-locate with other activities that require a similar set of inputs [9–14]. While there is an extensive research programme in the traditional economic growth and development literature for understanding how specific inputs, such as natural resources, public goods, labour, physical and

human capital, institutions or amenities, determine the choices of workers and firms for where to locate and what to produce [15], there is less research on more agnostic models that do not specify *a priori* what those inputs are. The latter has been a big part of the research programme in the interdisciplinary field of economic complexity [6], and several dimensionality reduction algorithms have been applied to data to extract 'metrics of complexity' which quantify the availability and sophistication of inputs present in an economy [2,6,16–22].

The complexity metrics computed from these algorithms were originally developed to explain differences in wealth and trade patterns across countries [2,17,23], and have since been shown to be highly correlated with both the levels and the change of aggregate economic output at higher geographical resolutions, such as regions and cities. For example, economic complexity metrics were recently applied to US regions and cities [24], prefectures in Japan [25], provinces in China [26], Indian states [27], Mexican states [28] and Colombian cities [29], and shown to explain different aspects of their economic growth (see [6,30] for a review of the literature).

So far, however, these dimensionality reduction algorithms have been difficult to connect with theoretical models of how economies work (see [31,32]). That is, there is no assurance that the manipulations of the data that these algorithms conduct actually quantify what they claim to quantify: the number of capabilities available in an economy, or required by an economic activity. For this reason, their use and interpretation has remained highly contested [21,33,34]. Ensuring that the mathematical foundations of economic complexity metrics are robust, that their method of estimation is reliable, and that they stand as meaningful economic indicators is key to understanding cities as complex systems, and acting upon them as such. In this work, we present a novel first-principles model that aims to address these shortcomings. That is, we present how a simple mathematical model can characterize the way inputs combine in cities to generate output (without making strong assumptions about what these inputs may be), how complexity variables then emerge and a method to estimate them, and how they correlate with measures of urban economic performance.

# 2. Model of urban economic complexity

The model we propose seeks to understand the production of units of output in cities. For illustration purposes, we will consider employment as the output of interest, but other forms of production such as the creation of firms, the production of patents, or occurrence of crime, could have been considered as well.

In this context, we propose a 'production recipes' approach to understand the patterns of employment probabilistically [35]. This approach suggests that questions like *why does a given industry employ more workers* per capita *in some cities than in others?* or *why in a given city do some industries employ more workers than others?* should be framed as questions about the mathematical shape of the urban production function across different types of phenomena. The production recipes approach is different from traditional economic models because it assumes that the statistics observed in economic data are more a function of how inputs combine rather than what those inputs are. The simplicity of this probabilistic approach emphasizes that urban production processes go beyond economic processes, which is why it can include phenomena like crime and disease.

Several models and explanations have been proposed to make sense of the differential shares of employment across industries. We argue that our proposed model offers a simple alternative that, for example, does not need assumptions about prices and markets to account for the empirical patterns observed (see [36] for a review of the literature). Two empirical patterns suggest that the mechanisms go beyond economics, and thus support our choice for a production recipes approach: the matrix of cities and industry presence displays 'nestedness' [37], and the employment is associated with a superlinear function of population size [38]. That is, 'nestedness' whereby larger cities have most industries (including those present in the least diversified cities), and urban scaling whereby larger cities have, on average, more absolute employment in any given industry, relative to small cities. The increase in employment $Y_{c,f}$ in city $c$ in an industry $f$ with city size follows a power-law function, $Y_{c,f} \propto N_c^\beta$ [38]. The nestedness and the power-law are also observed in other urban phenomena such as cases of crime, infectious disease prevalence, educational attainment and technological innovation [7,39–45]. The nestedness and the power-law relationships across such a diverse set of phenomena beyond economics are both key empirical observations that suggest that a general mechanism must be at work (see [46,47] for Ricardian models attempting to explain the nestedness). In §§3.1 and 3.2, we show how scaling laws and nestedness emerge from the model we propose.

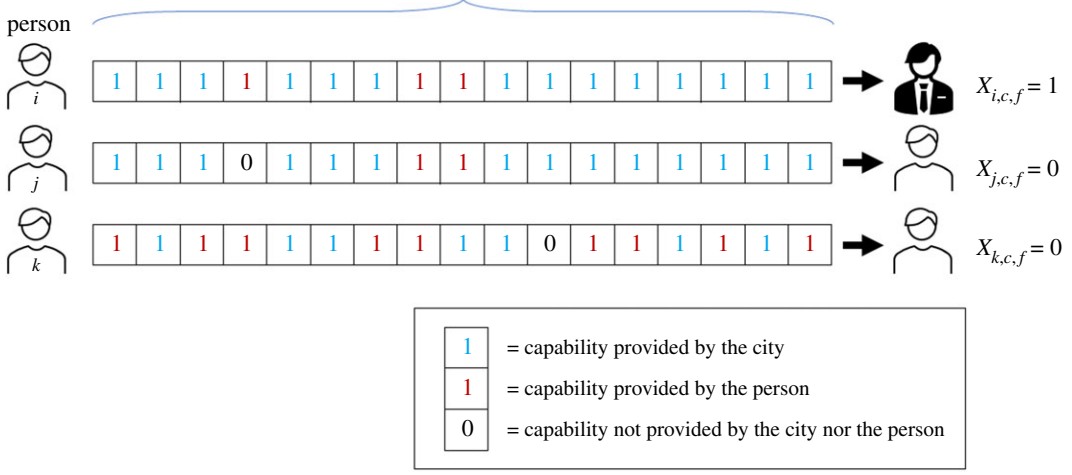

**Figure 1.** Sketch of the model. Three different individuals $i$, $j$ and $k$ in city $c$ and activity $f$. Individuals get their capabilities to work in $f$ either from their own effort (red 1s) or from the city (blue 1s). Note that individuals may differ in the way they get the capabilities. Only individual $i$ is able to get a job at activity $f$ in the city. Individuals $j$ and $k$ fail to get employed because they failed to get one of the capabilities. We quantify the number of factors required by activity $f$ by $M_f$, the likelihood of drawing a 1 from city $c$ by $r_c$, and the likelihood of drawing a 1 by the $i$th individual by $s_i$.

The theory we apply here is based on three assumptions. First, industries are defined as different conjunctions of complementary factors, as it has been shown in [48]. Complementarity implies that, for a unit of employment to be created in an industry, all factors must be simultaneously present. More complex industries are those that require more underlying causal factors. Second, cities possess a multiplicity of these factors.[1] And third, each person in the city has an (individual) ability to get their own factors. In general, we refer to these factors as 'capabilities', and we use these terms interchangeably. We claim that the model reveals a simple methodology to estimate these underlying complexity metrics that allows us to understand the differences in average wages and average size of establishments across cities and industries.

Since the model is built upon the concept of a 'capability', it is worth briefly discussing what capabilities stand for. The idea is that capabilities are inputs of production, but getting the right capabilities may require different instances of knowledge. Hence, a capability can be acquired either by know-what (e.g. knowing which specific component is needed for a machine to operate), know-why (knowing the causal mechanisms that are critical for the machine to work), know-who (knowing who has a specific expertise to fill in a specific technique), and so on. In general, however, there are reasons to believe that the most determinant component of knowledge is tacit [49–51]. This type of knowledge is generally referred to as 'know-how'. Know-how can be possessed by the individual in question, or distributed in the city and accessed by living in it and interacting with its infrastructure and its citizens [51]. This is the reason we use 'know-how' as part of the description of the variables derived from the model. In our model, the person-specific probability of possessing any one capability ($s_i$) may be associated with occupation-specific and industry-specific know-how carried by individuals, while the city-specific probability of possessing any one capability ($r_c$) may be associated with distributed know-how, specific to a location, collectively possessed by the city (see, however, [52] for a study in which each of these types of knowledge is claimed to be internalized by workers and shown to affect patterns of economic diversification; for a review of the factors affecting diversification see [53]).

Figure 1 presents a conceptual sketch of how our model works and shows that production is the process of combining a multiplicity of complementary inputs (for similar models, see [2,54–56]). In the model, individuals will be indexed with the letter $i$, cities with $c$ and industries with $f$. With this set-up and the conceptual sketch in figure 1, three fundamental variables emerge from the model:

1. $M_f$: This is the number of capabilities required to produce the industry-specific product $f$. We can refer to it as the intrinsic 'complexity' of the industry.

---

[1]In [7], we developed a model in which factors are acquired through a stochastic process of accumulation, but here we simply take the factors present in a city as a given.

2. $s_i$: This is the person-specific probability of possessing any one capability. Accordingly, $s_i$ can be referred to as a measure of 'individual know-how'.

3. $r_c$: This is the city-specific probability of possessing any one capability. Accordingly, $r_c$ can be referred to as a measure of 'collective know-how' and represents a measure of input availability in the urban environment of city $c$.

Importantly, these parameters will be estimated from data, rather than estimated based on ad hoc definitions. Specifically, using cross-sectional data on employment across cities and industries, we will make inferences about the current value of these hidden variables.

To derive a connection between data and these variables, we need to understand how, in this model, the recombination of complementary capabilities in cities gives rise to probabilities of employment. To that end, let the random variable $X$ be equal to one if the person is employed and zero if not. We will derive the probability, $p$, that someone is employed in a city in an industry as a function of the complexity of the industry complexity $M_f$, the person's individual know-how $s_i$ and the city's collective knowhow $r_c$. We will not, however, model the dynamics of $M_f$, $s_i$, or $r_c$. In particular, introducing laws of agglomeration or congestion are left out for future research (see electronic supplementary material, B for a brief discussion).

The parameter $M_f$ represents the 'inherent complexity' of the economic activity associated with the production of industry's product $f$. The more capabilities are needed, the larger the value of $M_f$ and the more complex the activity. We emphasize that the model is fully agnostic as to what capabilities are involved in these activities. The complexity $M_f$ of an industry may not necessarily be associated with the number of employees required by a typical firm, nor with the level of schooling of the workers employed in $f$ (e.g. industries that tend to employ people with PhDs versus industries that employ low-skilled workers). Thus, $M_f$ may ultimately quantify something intangible and difficult to measure directly in the real world. For example, industries that use low human capital but that are intensive in physical capital may have high $M_f$ if they require the coordination of a large set of inputs and materials to be transformed with the use of sophisticated machinery in a complex production process.

The parameter $s_i$ represents how many capabilities the individual $i$ can acquire on their own. Specifically, it is the probability that she can get *any* of the $M_f$ capabilities required by the typical business in industry $f$. This probability can be interpreted as a measure of their individual know-how. Since capabilities are supposed to vary qualitatively from one another, $s_i$ is meant to capture the *breadth* rather than the *depth* of knowledge. It is about how many different things they could know how to do by themselves. The explicit inclusion of this parameter in the model represents an important contribution to the framework of economic complexity (cf. [32]). Since capabilities represent units of knowledge that are interconnected, depth and breadth of knowledge (as measured by their level of schooling, or by the different things they know how to do, respectively) may be both positively correlated with $s_i$.

The parameter $r_c$ models the idea that the city provides workers with a variety of capabilities through exposure to other people, services and institutions within the urban milieu. Presumably, the bigger the city, the more diverse and the more capabilities it can offer. Thus, $r_c$ is the *probability* that the city $c$ provides *any* capability.[2]

By solving the model, we get that the probability that individual $i$ will be employed in industry $f$ given that they live in city $c$ can be written as

$$\Pr\{X_{i,c,f} = 1\} = e^{-M_f(1-s_i)(1-r_c)}. \tag{2.1}$$

See electronic supplementary material, A for details on the derivation of equation (2.1).

Our model shares many similarities with the O-ring model of production [54]. As in the O-ring model, for example, employment is guaranteed only if all required capabilities (or 'tasks') are combined. Our model differs from it, however, in that any given capability required by a citizen to work in a industry can be provided by the worker herself or, alternatively, they can take it from the city if available. This framework allows us to see clearly that urban phenomena occur because

---

[2]We can imagine that the city has a 'field' spread in the $xy$-coordinates, $r_c(x, y)$. This field is an abstraction of the urban milieu that represents the location-specific probability that the city provides one of the ingredients for phenomena to occur, and we can assume that people interact with it as they live and work in the city. It should capture the elements from all the types of urban interactions to which people are exposed through the social, economic and built environment. In this view, the city functions as though it is permeated across space by a 'cultural field', and $r_c(x, y)$ quantifies the magnitude of the social, economic and cultural repertoire available at a particular location. Locations where the value of the field is high have a high concentration of diverse urban factors.

individuals are able to 'execute' a recipe (e.g. a production process, a programme or algorithm) if the environment is favourable, that is, if the city complements the individual. How complex a given recipe is, how capable an individual is, and how suitable the city is for executing the recipe, are the three fundamental quantities that determine the overall statistics of employment in cities, as well as other measures of urban output.[3]

# 3. Theoretical and methodological implications of the model

## 3.1. Urban scaling laws

Equipped with equation (2.1), we can reproduce the power-law relationships of employment with population size (see [38]). To that end, we invoke models of cultural evolution which predict a specific association between population size and the size of collective know-how, $r_c = a + b\ln(N_c)$ (see [61–63]). This relation is consistent with observations in urban contexts, in which the diversity of factors and capabilities in city $c$ is approximately a logarithmic function of population size [38]. Incorporating this relationship between $r_c$ and $N_c$ into equation (2.1) yields $p_{i,c,f} = e^{-M_f(1-s_i)(1-a)} e^{bM_f(1-s_i)\ln(N_c)} = e^{-M_f(1-s_i)(1-a)} N_c^{bM_f(1-s_i)}$. It becomes clear why a power-law function emerges, since we are exponentiating a logarithm (see [7] for details about testing the predictions of this explanation). This result indicates that more complex phenomena scale more superlinearly, explaining why more complex technological innovations and production processes tend to occur in larger and more diverse urban hubs [7,8,46,64]. Relatedly, it also explains why and how different phenomena of the same kind (e.g. two sexually transmitted diseases, two types of college degrees or two types of crimes), which presumably are driven by similar networks of social interaction, can feature very different urban scaling patterns (see [43,44,65,66]).

Although these are solid theoretical grounds on which to base the relationship between $r_c$ and $N_c$, below we will estimate directly the parameter $r_c$ from data, and show how its association with the logarithm of $N_c$ also emerges as an empirical result.

## 3.2. Nestedness of activities across locations

The nested (triangular) pattern of industries across cities (e.g. [37]) emerges naturally from the function in equation (2.1). To show that, assume for simplicity that $s_i$ is approximately constant for all individuals in all locations. Next, let there be two industries $L$ and $H$ such that $M_L < M_H$, and two cities $l$ and $h$ such that $r_l < r_h$. According to equation (2.1), the probability of employment is higher in the city with high collective know-how in both industries, since $p(M_H, s, r_h) > p(M_H, s, r_l)$ and $p(M_L, s, r_h) > p(M_L, s, r_l)$. In addition, the share of employment in high-complexity industries with respect to low-complexity industries in diverse cities will be larger than the same ratio in less-diverse cities, since $p(M_H, s, r_h)/p(M_L, s, r_h) > p(M_H, s, r_l)/p(M_L, s, r_l)$. The latter property is called 'log-supermodularity', and reflects why diverse cities are disproportionately more competitive in complex economic activities than less-diverse cities [8,19,46,67,68]. This result is shown graphically in figure 2, where cities and industries have been given different values of $r$ (distributed uniformly between 0.7 and 0.95) and $M$ (distributed uniformly between 5 and 25), and $p(M, s, r)$ (for constant $s = 0.1$) is computed. The triangular pattern that emerges shows that probabilities of employment are nested, as empirically observed [37]. According to the model, it is clear that this nested pattern can become more drastic if we assume individuals with high individual know-how (high $s$) tend to locate themselves in cities with high $r$. This sorting of individuals is observed empirically and we discuss in the next section how it may emerge from the model.

---

[3]A note about terminology may be necessary here. All three measures increase in value with the number of distinct capabilities considered. Hence, they can be thought of as measures of diversity (or variety, as defined by [57]; see also [58]). The more capabilities are required by an industry, or possessed by an individual or by a city, the more possibilities of recombining capabilities there are (for example, $r_c$ itself has the potential to create a constant push towards greater $r_c$ as argued in electronic supplementary material, C). This aspect of the model supports the interpretation of these quantities as measures of complexity as well (in the same vein that [59] characterizes how biological diversity and complexity go hand-in-hand), despite the fact that diversity and complexity are typically considered to measure different aspects of a system (for a thorough review of these concepts see [60]).

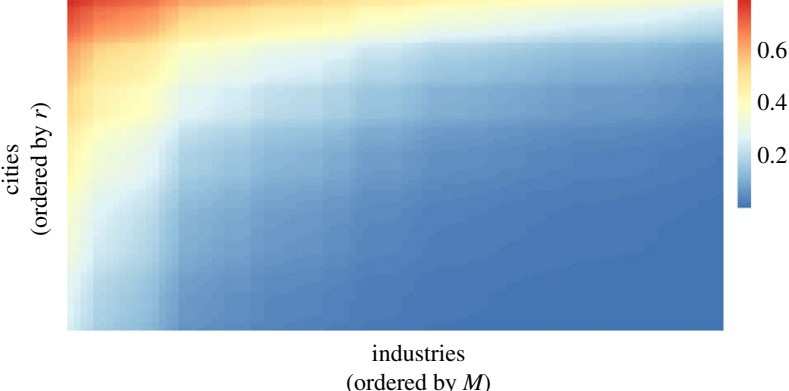

**Figure 2.** Nestedness emerges from the model. For 100 cities and 100 industries, the probability of employment is shown, with $s = 0.1$ assumed constant, and with $r$ distributed uniformly between 0.7 and 0.95, and $M$ distributed uniformly between 5 and 25.

## 3.3. Sorting of individuals with high know-how

Here we show some arguments that suggest the model may reproduce the observation that skilled individuals would tend to locate themselves in large diverse cities [69]. This is shown by noting that the probability in equation (2.1) implies a form of assortative matching between individuals and cities, assuming an aggregate maximization of total employment across all cities subject to congestion costs (see details in electronic supplementary material, B). To show this, let there be two individuals $L$ and $H$ such that $s_L < s_H$ and two cities $l$ and $h$ such that $r_l < r_h$. The question is how will the two individuals sort themselves into these two cities, assuming there is a cost for being in the same city. Given this, we have that $p(M, s_L, r_l) + p(M, s_H, r_h) > p(M, s_L, r_h) + p(M, s_H, r_l)$, for a fixed (but large) $M_f = M$ (see electronic supplementary material, B). In words, the most likely situation is one in which the individual with high (low) levels of know-how will sort to places with high (low) levels of collective know-how. In a more complete model with prices and better specification of the costs of congestion, the phenomenon of assortative matching should ultimately be reflected in wages, as has been observed [69,70]. We will explore this issue empirically by studying how our variables of economic complexity, once estimated, correlate with wages in cities across industries.

## 3.4. Relationships between the three drivers of urban economic complexity

We refer to the (negative) exponent in equation (2.1) as a 'net complexity' (see [71]). This exponent is decomposable into the original three drivers of urban complexity. They are, *a priori*, independent quantities.[4] Presently, we lack a detailed theory about the dynamical laws of the variables $M_f$, $s_i$ and $r_c$ within our modelling framework, and how they relate to one another. Still, it is reasonable to expect that increases in collective know-how (through immigration that brings new capabilities and know-how,[5] direct foreign investments that inject specific capabilities to specific industries, or by innovation) may have a reinforcing effect that sparks and fuels a virtuous cycle: a place with a relative large enough collective know-how will attract more people and facilitate the appearance of more complex economic activities, which will increase in turn the collective know-how in that place. This process will thus propel a runaway cycle of collective learning that will concentrate economic activities and wealth in large cities (e.g. [29]). The more complex the activities, the more concentrated they will be in fewer places. Interestingly, changes in these three terms have *exponential* effects in the probability of employment (more details in electronic supplementary material, C).

The fact that the net complexity term is decomposable leads to a strategy for estimating its components by taking logarithms twice, converting an exponential of a product into a simple sum. This method is valid only if our model is an accurate description of how employment is created. To test the latter assumption, in what follows, we will estimate these quantities and demonstrate that

[4]Statistically, however, we expect them to be correlated, since firms with complex production processes (high $M_f$) are likely to choose to locate in diverse cities (high $r_c$), which are the places where high-skill individuals (high $s_i$) sort themselves into (see electronic supplementary material, B for details).

[5]Models such as that in [72] show that when skills of immigrants are complementary to those of locals, the wages of both locals and immigrants increase.

equation (2.1) provides important statistical improvements over alternative conceptualizations from both modelling and predictive power perspectives. Crucially, the resulting estimates of the city-specific driver (urban collective know-how) is associated with measures of economic performance. In brief, we show that the 'drivers' of urban economic complexity can be measured and have measurable consequences.

# 4. Material and methods

## 4.1. Data

We use data on the estimated counts of employment, number of establishments and average wages by city-industry-year. Data were downloaded using the programming codes made available by the Bureau of Labor Statistics. See electronic supplementary material, D for additional details of the data.

## 4.2. Estimating the drivers of urban complexity from data

Our estimates of complexity (for industries, cities and individuals) stand for parameters in our mathematical model. In this section, we show how to estimate these three drivers of complexity using estimated fixed effects in a regression.

Assume, for now, we know the value of the probability in equation (2.1), $p_{i,c,f}$. Equating such estimate to the proposed functional form, and taking negative logarithms twice, yields

$$-\ln(-\ln(p_{i,c,f})) = -\ln(M_f) - \ln(1 - s_i) - \ln(1 - r_c). \tag{4.1}$$

Equation (4.1) shows how net complexity is decomposed linearly into its main components. To estimate the value of the economic complexity variables in our model, equation (4.1) implies we can regress $-\ln(-\ln(p_{i,c,f}))$ against three *additive* fixed-level effects corresponding to the activity, the individual and the city.

The regression method we propose, however, has some limitations in practice. As it is typically the case, microdata at the individual level is often difficult to obtain, which makes $p_{i,c,f}$ difficult to estimate. When the only information available is aggregate counts of employment per industry and city, we cannot estimate the probability on the left-hand side of equation (4.1) at the level of each individual, and therefore we cannot include an individual-fixed effect on the right-hand side.

Fortunately, the model is simple enough that we can address this limitation by substituting $s_i$ in equation (2.1) with the *average* individual know-how in city $c$, $\bar{s}_c \approx \sum_{i \in c} s_i / N_c$. This substitution implies that $\Pr\{X_{c,f} = 1\} = \exp(-M_f (1 - \bar{s}_c)(1 - r_c))$ (see electronic supplementary material, E). This expression represents the probability that *any* random individual in city $c$ (as opposed to a specific one) is employed in activity $f$. We estimate $p_{c,f} = \Pr\{X_{c,f} = 1\}$ through the share of employment $\widehat{p}_{c,f} = y_{c,f} = Y_{c,f}/N_c$, where $Y_{c,f}$ is the employment count in industry $f$ in city $c$ and $N_c$ is total population size.[6] The regression equation that will allow us to estimate the complexity of industries and the collective know-how of cities (but not the individual know-how of workers) becomes

$$-\ln(-\ln(y_{c,f})) = \delta_f + \gamma_c + \varepsilon_{c,f}, \tag{4.2}$$

where $\varepsilon_{c,f}$ is the error term, $\delta_f$ stands for $-\ln(M_f)$ and $\gamma_c$ for $-\ln((1 - \bar{s}_c)(1 - r_c))$.[7] We note that the city fixed effect is a city-specific variable that includes the interaction between the suitability of the urban environment and the average capacity of citizens. While we are unable to learn much about individual-level know-how from our data, the inclusion of a proxy allows us to learn about the effect that the other two drivers have on measures of urban performance (§4.4).

---

[6]The estimate of $p_{c,f}$ could in principle incorporate a Bayesian prior using pseudocounts, $\widehat{p}_{c,f} = (Y_{c,f} + \alpha_{c,f})/(N_c + \sum_f \alpha_{c,f})$ in order to handle the case when $Y_{c,f} = 0$. See [73].

[7]If one has an estimate of $\widehat{p}_{c,f}$ for all possible combinations of $c$ and $f$ in the data (i.e. there are no missing values in the matrix of places and activities), the method of estimation is even simpler. Applying a singular value decomposition (SVD), or principal component analysis (PCA), to $\log(\widehat{p}_{c,f}) = -M_f (1 - \bar{s}_c)(1 - r_c)$, will yield a single pair of vectors, one vector of principal components and another of principal loadings, corresponding to the vector of $(1 - \bar{s}_c)(1 - r_c)$ and $M_f$, respectively. Since missing values in data are more the rule than the exception, the method of fixed effects proposed in this paper is likely to have greater applicability than the application of SVD.

## 4.3. Evaluation against competing models

We evaluate the predictive power of model (4.2) with respect to four alternative models using holdout data. Two of the models differ from our model only in terms of the functional transformation of the dependent variable; that is, rather than $-\ln(-\ln(y_{c,f,t}))$, the left-hand side is given by $y_{c,f,t}$ (namely Model 1.1) and $\ln(y_{c,f,t})$ (Model 1.2), respectively. As shown in electronic supplementary material, F, Model 1.2, for example, is derived from a world in which capabilities are not complementary (as we have assumed) but substitutable. As a consequence, differences in predictive performance between model (4.2) and these two other models will inform us about the importance of the specific functional form predicted by the theory and its underlying assumptions.

The other two alternative models differ from (4.2) in the formulation of the right-hand side of the model. One model (namely Model 2.1) will be the standard urban scaling model where it is assumed that the scaling exponent is the *same* for all phenomena, as suggested by network-based explanations [74]. The last alternative model (Model 2.2) is an unconstrained version of the standard scaling model, where it is assumed that both the baseline prevalence and the scaling exponent differ, in principle, for each industry $f$ (see electronic supplementary material, F). Differences in performance between model (4.2) and these last two models will inform us about the validity of adding degrees of freedom to explain employment patterns across cities and industries.

To compare these models based on 'out-of-sample' prediction performance, for each year, we split the data into training and testing (or validation) sets. After the parameters of the models are fitted using the training data, they are then compared by how accurately they predict the dependent variable on the test set. The predictions were evaluated using the root mean squared error (*RMSE*) and the mean absolute error (*MAE*). The train and test random splits were repeated 100 different times of the data (bootstrapping cross-validation). See electronic supplementary material, F for more details.

## 4.4. Linking the drivers with urban economic performance

There is a tension between individuals and firms trying to reap the benefits of diversity in cities, and individuals and firms trying to avoid each other to reduce congestion costs from living and operating in the same city. As we argued in §3.3, however, we expect skilled individuals and complex industries to locate themselves in large cities, paying higher wages to compensate for the costs of congestion. Thus, in the empirical analysis that we propose in this section, we are interested in analysing whether our variables of interest (industry complexity and collective know-how) are *positively* associated with measures of economic performance of firms and workers. We will measure performance with average wages and average size of establishments at the level of city–industry combinations.[8]

Given that firms and workers in larger (more populous) cities are more productive [75–85], we also included in our regressions the logarithm of city population size as a control variable. In addition, we will include two measures of city and industry complexity that have been proposed before in the literature on economic complexity. These measures are known as the economic complexity index (ECI) and the product complexity index (PCI) [2], here applied to cities and industries (instead of countries and export products).[9] These indices are not statistical estimates of know-how or complexity. Instead, they are a mathematical construction, which, given their empirical high correlation with measures of economic growth [24], is assumed to rank cities and industries by their underlying number of capabilities (see [21] for a discussion, and [31] for analytic justifications of this method).

We will analyse these associations through linear regressions of the following form:

$$\ln(z_{c,f,t}) = \beta_0 + \beta_1 \ln(\bar{s}_{c,t}^{\text{proxy}}) - \beta_2 \widehat{\delta_{f,t}} + \beta_3 \widehat{\gamma_{c,t}} + \mathbf{X}_{\text{controls}} \boldsymbol{\beta}_{\text{controls}} + \varepsilon_{c,f,t}. \tag{4.3}$$

where $z_{c,f,t}$ stands for either wages or size of establishments, and where we have made explicit the time dimension $t$ representing years. We will carry out different specifications of regression model (4.3) for different combinations of the explanatory variables.[10] In the main text, we will show the results for

---

[8]Here, we face one challenge regarding what the data measure. The Bureau of Labor Statistics provides data about establishments, not firms, at the level of city–industry combinations. Hence, a large firm may be composed of many small establishments, and these may affect the validity of our regression model equation (4.3).

[9]These indices are based on a spectral clustering method [86–88]. As such, the method provides one vector of ratings for cities and another for industries, and these ratings cluster cities and industries according to their pairwise similarities deduced from the presence/absence matrix of industries across cities.

[10]For simplicity of presentation, we have not included diagnoses of spatial autocorrelations. However, researchers seeking to explain urban outcomes should take these into consideration, since cities are not independent of one another.

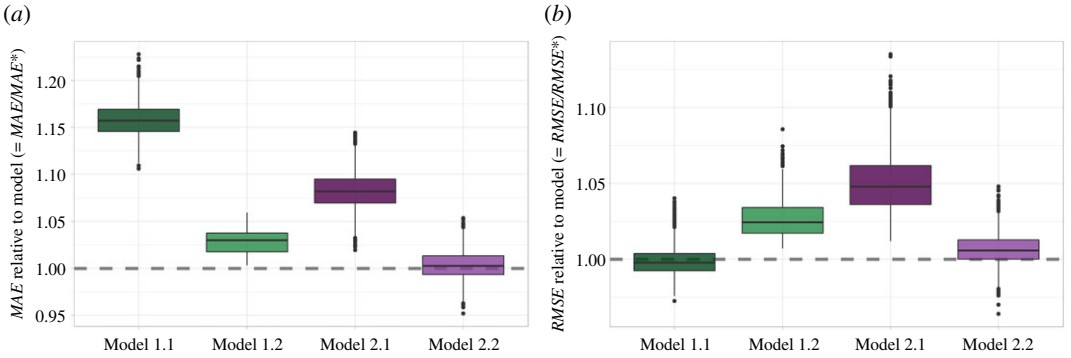

**Figure 3.** Evaluation against competing models. Comparison of out-of-sample predictions from all models using 100 random cross-validation train/test splits. (a) y-axis represents the ratio between the mean absolute error (*MAE*) of each alternative model divided by the *MAE*\* of model (4.2). (b) y-axis represents the ratio between the root mean square error (*RMSE*) of each alternative model divided by the *RMSE*\* of model. For both (a) and (b), values larger than one represent worse out-of-sample predictions as compared with our model.

one specific year, but in electronic supplementary material, L, we show the results for all years, conducting one regression for each year separately to control for changes in nominal prices.

As a consequence of lacking individual-level microdata, and as explained in §4.2, we do not have an estimate of individual know-how that we can include in equation (4.3). However, we reduce the potential effects of omitted variable bias by including a quantity that acts as a proxy for individual know-how in equation (4.3), $\ln(\bar{s}_{c,t}^{\text{proxy}})$. For this, we consider educational attainment. More years of educational attainment are usually a measure of specialization, but they can also be an indication of a person's competency to perform several productive tasks, as our model assumes. Whether years of schooling is a good proxy for individual know-how, however, remains an empirical question (see, e.g. [89]). Acknowledging these caveats, we will proxy $\bar{s}_{c,t}$ using the average levels of schooling in the city $c$ in year $t$ (see electronic supplementary material, E for details). Thus, statements and hypotheses about how, based on our model, this driver affects wages and establishment sizes should be taken with caution.

Finally, we note that we use as independent variables in equation (4.3) the estimates of industry complexity and collective know-how, $-\widehat{\delta_{f,t}}$ and $\widehat{\gamma_{c,t}}$, coming from the results of §4.2. Moreover, we note the use of a negative for $\widehat{\delta_{f,t}}$ and a positive one for $\widehat{\gamma_{c,t}}$ such that their interpretation is consistent with measures of complexity and know-how.

# 5. Results

## 5.1. Comparing prediction power of models

Figure 3 presents a comparison of our model (4.2) against four alternative models, as specified in §4.3. To make the comparisons clearer, in the figure, we have divided the *MAE* score (and *RMSE*) of each alternative model by the *MAE*\* (*RMSE*\*) of our model in each run of the bootstrapping cross-validations. Models with worse predictive performance than ours will be above the dashed line.

Results show that our model has superior performance in terms of *MAE* except for the unconstrained scaling model (Model 2.2, where baseline prevalence and the scaling exponent can differ for each industry $f$). The comparable performance of Model 2.2 is supportive of the ideas and results put forth by [7]. Our model has also superior performance in terms of *RMSE* with respect to all models except for the first alternative model (Model 1.1, where the dependent variable is not logged). Interestingly, this indicates that city and industry fixed effects directly fitted on *per capita* employment provide an alternative good fit to the data, but only when there are no extreme values (*RMSE* is more sensitive to outliers).

## 5.2. Relation of collective know-how to population size and industrial diversity

Having estimated $\gamma_{c,t}$ and $\delta_{c,t}$, as described in §2.1, we present a descriptive understanding of how much these estimates conform to our proposed notions of collective know-how and industry complexity.

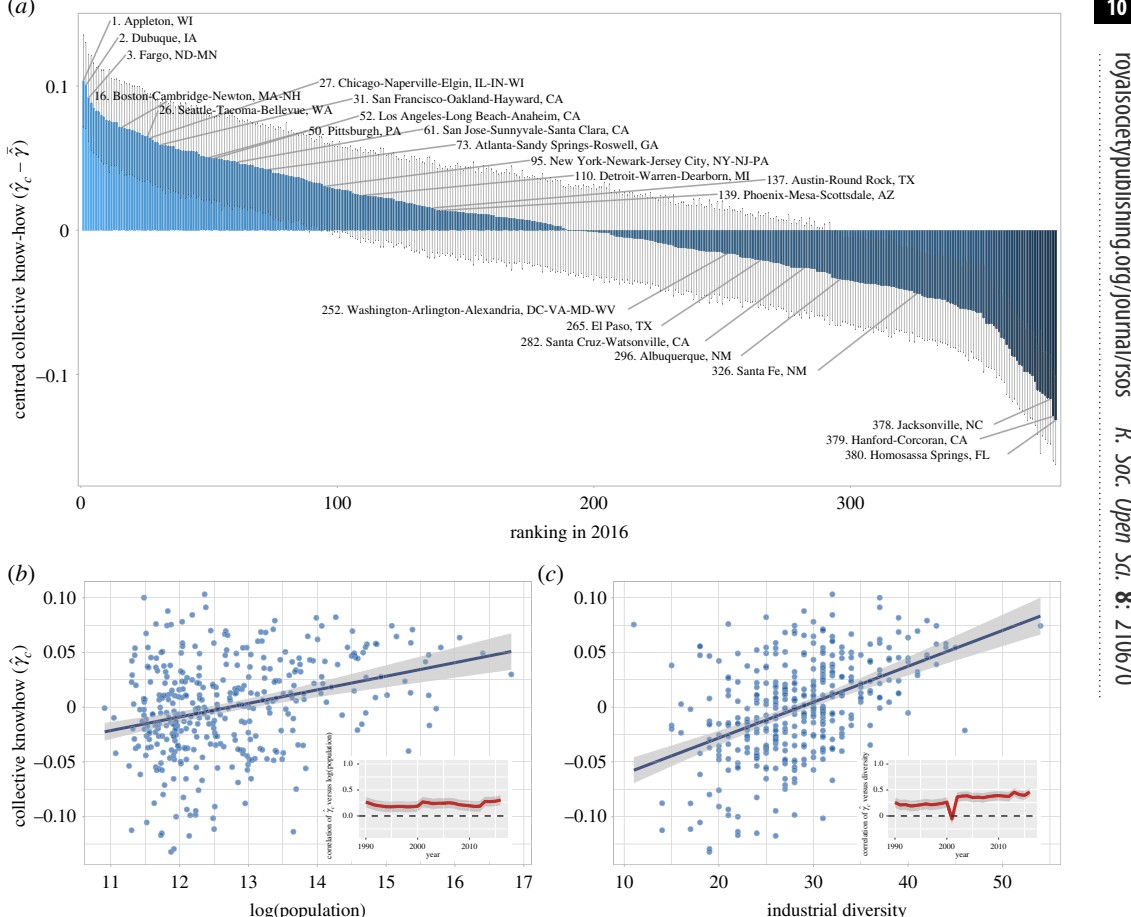

**Figure 4.** Estimated collective know-how across US cities. (*a*) Bar plot and the ranking of cities according to their centred scores of collective know-how in 2016. The plot shows ± 1 s.e. coming from the regression estimation for every city. (*b*) Association with (log) population size in 2016 (Pearson $\rho = 0.30$, $R^2 = 0.09$). The inset shows the Pearson correlation across years. (*c*) Association with industrial diversity in 2016 (Pearson $\rho = 0.46$, $R^2 = 0.21$). The inset shows the Pearson correlation across years.

Figure 4*a* shows that most large cities have scores of collective know-how that are above the average. Among the large cities, Boston has the largest score. One exception is the metropolitan area of Washington, DC, which falls below the average. Interestingly, New York City is closer to cities like Detroit, Austin and Phoenix, than to cities like Chicago, Los Angeles or Boston. The estimates of collective know-how, however, have large standard errors (shown as grey segments in each bar) due to the fact that the estimation of the fixed effect relies on few industries present per city (30 industries per city on average).

Figure 4*b*,*c* shows how the scores correlate with population size and industrial diversity, respectively. Here, we define industrial diversity by the number of industries in each city that have location quotients larger than one. That is, if $LQ_{c,f} = (Y_{c,f}/\sum_f Y_{c,f})/(\sum_c Y_{c,f}/\sum_{c,f} Y_{c,f})$, then diversity is $d_c = \sum_f \mathbf{1}_{(LQ_{c,f}>1)}$ (e.g. see [2]). Our estimate of $\widehat{\delta_{c,t}}$ increases with both population size and industrial diversity, supporting the assumption that it fits the pattern of a measure of collective know-how. The association with diversity, interestingly, has been increasing with time (see inset of figure 4*c*), which may suggest the fact that the process of diversification is driven by larger bodies of collective know-how.

## 5.3. Relation of industry complexity to occupational diversity and geographical ubiquity

Next, we assess whether our estimate of the complexity of industries is consistent with our notion of 'difficulty'. Thus, we compare our estimates with a measure of occupational diversity on the one hand, and with a measure of geographical concentration on the other. We define 'occupational effective diversity' based on the shares of employment across occupations per industry, computing it by taking the exponential of Shannon's entropy. That is, if $p_{f,o} = E_{f,o}/\sum_o E_{f,o}$ is the US national share of employment of occupation $o$ in industry $f$, then the occupational effective diversity is $d_f = \exp\{-\sum_o p_{f,o}\ln(p_{f,o})\}$ (see [90–92], and electronic supplementary material, D for the source of

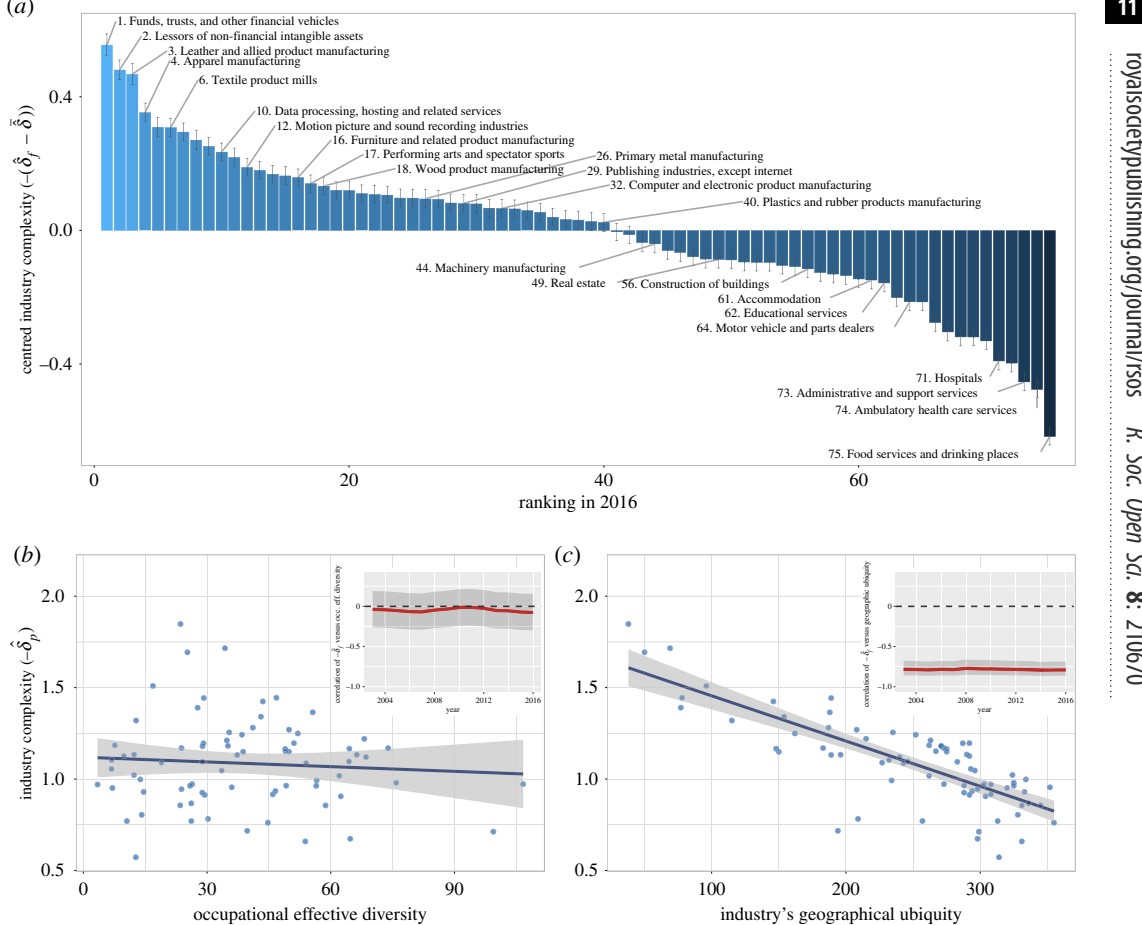

**Figure 5.** Estimated industry complexity across three-digit NAICS industries. (*a*) Bar plot and the ranking of industries according to their centred scores of industry complexity in 2016. We note that we show the negative sign next to $\hat{\delta}_f$ so that higher levels are interpreted as higher complexities. The plot shows $\pm$ one standard errors coming from the regression estimation for every industry. (*b*) Association with the effective number of occupations per industry in 2016 (Pearson $\rho = -0.08$, $R^2 = 0.01$). The inset shows the Pearson correlation across years. (*c*) Association with the geographical ubiquity per industry in 2016 (Pearson $\rho = -0.79$, $R^2 = 0.62$). The inset shows the Pearson correlation across years.

occupation-industry data). The measure of geographical concentration, 'ubiquity', is analogous to above's measure of industrial diversity for cities. That is, $u_f = \sum_c \mathbf{1}_{(LQ_{cf} > 1)}$ (e.g. see [2]).

Figure 5*a* shows that many financial and manufacturing-related industries have scores that are above the average. Interestingly, 'Performing arts and spectator sports' appears in ranking 17, above industries like 'Computer and electronics' or 'Plastics and rubber products' manufacturing. The least complex industries, not unsurprisingly, are some common service industries, with 'Food services and drinking places' at the bottom of all.

Figure 5*b,c* shows how the scores correlate with the measure of occupational diversity of industries, and with the measure of geographical ubiquity, respectively. Somewhat unexpectedly, the measure of complexity does not correlate with the effective number of occupations that are typically employed by the industry. This result may be a consequence of our model not taking into account the other economic forces at play, but it may also suggest that occupations are not the fundamental units of know-how. Conversely, the fact that the most complex industries occur in few metropolitan areas indicates that these industries are indeed dependent on the right urban context.

## 5.4. Relationship between the drivers and the establishment sizes and wages

Here, we show the associations of our estimates with wages and establishment sizes at two levels of aggregation: coarse (each observation is either a city or an industry) and fine-grained (each observation is a city–industry combination).

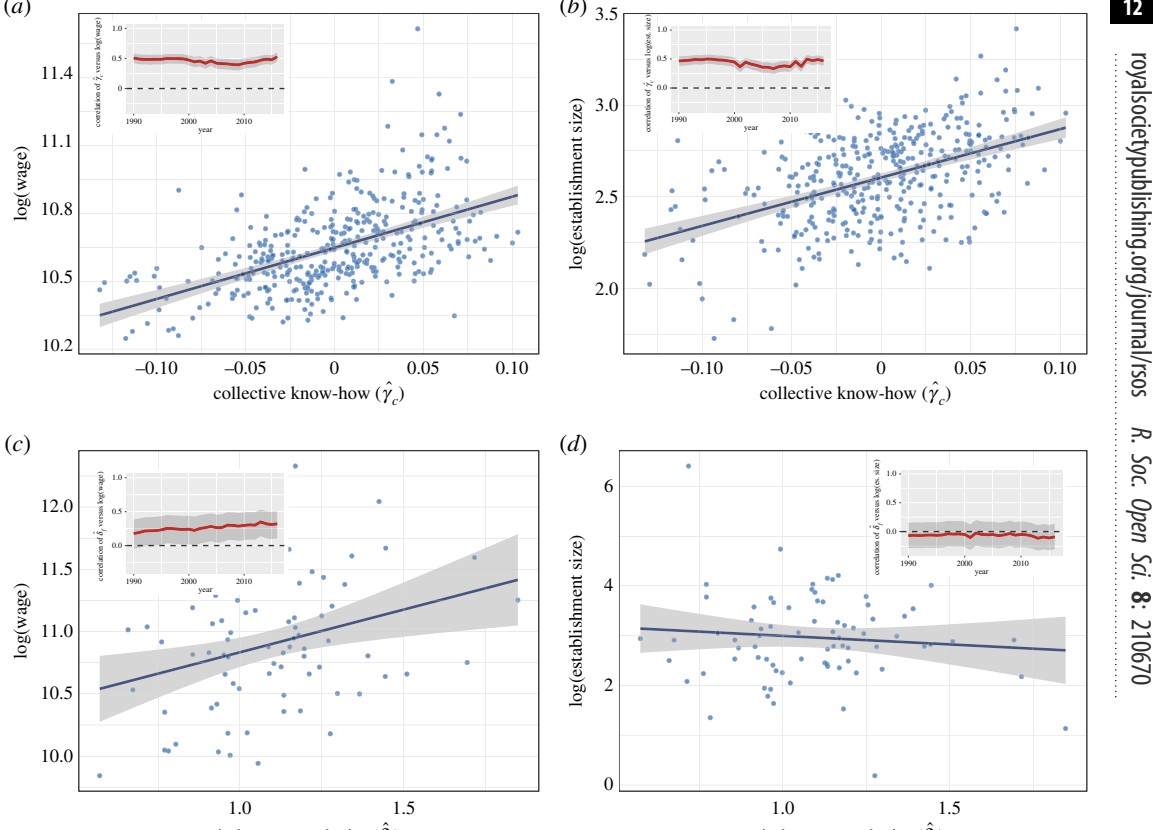

**Figure 6.** (a) Association of collective know-how with (log) average wages at the level of cities (Pearson $\rho = 0.53$, $R^2 = 0.28$). (b) Association of collective know-how with (log) average establishment size at the level of cities (Pearson $\rho = 0.47$, $R^2 = 0.22$). (c) Association of industry complexity with (log) average wages at the level of industries (Pearson $\rho = 0.32$, $R^2 = 0.10$). (c) Association of industry complexity with (log) average establishment size at the level of industries (Pearson $\rho = -0.09$, $R^2 = 0.01$). All scatter plots are for 2016 data, and all insets show the Pearson correlation across all years.

We start this section by showing the correlations when observations are aggregated at the city level and at the industry level, separately. These associations are shown in figure 6. In panels $a$ and $b$, each point is a city, and the figures indicate that the association between average wages and average establishment size with collective know-how is very strong. The association is stable across years, as shown in the respective insets. In panels $c$ and $d$, each point is an industry, and the figures show that the association of wages and establishment sizes at the aggregate level of industries, however, is less strongly associated with our estimate of industry complexity.

Next, we investigate the associations at a fine-grained level of aggregation in which each observation is a city–industry combination, as proposed in §4.4 in equation (4.3). At this level of disaggregation, there were 19 490 different city–industry combinations in our data in 2016 (out of $381 \times 78 = 29\,718$ possible combinations). The median number of employees per observation was 772 and the median number of establishments was 51. The question is whether our complexity variables can account for some of the variation in the average wages of workers and the average size of establishments across all these observations, and whether they remain statistically significant after adding other competing controls.

Tables 1 and 2 present the results of seven regression models applied to the year $t = 2016$, where the dependent variables are logarithm of establishment sizes and wages, respectively. The key observation is that the increase in the level of disaggregation makes the problem of predicting wages and establishment sizes more challenging (reflected in the low $R^2$ values), and yet, industry complexity and the city know-how remain statistically significant ($p < 0.05$) in the presence of other controls like population size, PCI and ECI.

In table 1, the coefficient for population size is fairly constant across specifications, indicating that in the cross-section of US metropolitan areas, a 1% increase in population size is, on average, associated with 0.13–0.15% increases in average establishment size. Average years of schooling is positively associated with establishment size, but when we control for the rest of the variables, it switches sign

**Table 1.** Associations with average size of establishments. Linear regressions for (log) average firm size at the city–industry level as a function of city (log) population size, (log) of average years of schooling, (log) of the inherent industry complexity, (log) of the collective know-how of city, controlling for economic and product complexity indices. Regression table shows only the year 2016.

| | Dependent variable: log(ave. establishment size) | | | | | | |
|---|---|---|---|---|---|---|---|
| | (1) | (2) | (3) | (4) | (5) | (6) | (7) |
| log(city population) | 0.152***  t = 24.865 | | | | | 0.140***  t = 22.224 | 0.131***  t = 17.029 |
| log(ave. yrs. schooling) | | 0.397***  t = 4.630 | | | −0.563***  t = −6.096 | −0.842***  t = −9.144 | −0.932***  t = −9.336 |
| log(industry complexity) | | | −0.852***  t = −24.610 | | −0.888***  t = −26.057 | −0.946***  t = −28.018 | −0.770***  t = −21.118 |
| log(city collective know-how) | | | | 3.750***  t = 23.947 | 4.387***  t = 25.639 | 3.539***  t = 20.429 | 3.324***  t = 17.861 |
| PCI | | | | | | | −0.025***  t = −12.654 |
| ECI | | | | | | | 0.031***  t = 3.005 |

(*Continued.*)

**Table 1.** (*Continued.*)

| | Dependent variable: log(ave. establishment size) | | | | | | |
|---|---|---|---|---|---|---|---|
| | (1) | (2) | (3) | (4) | (5) | (6) | (7) |
| constant | 0.691*** | 2.065*** | 2.644*** | 2.630*** | 3.445*** | 2.055*** | 2.317*** |
| | $t = 8.830$ | $t = 16.919$ | $t = 382.560$ | $t = 381.521$ | $t = 26.203$ | $t = 14.257$ | $t = 12.710$ |
| observations | 19 490 | 19 490 | 19 490 | 19 490 | 19 490 | 19 490 | 19 490 |
| $R^2$ | 0.031 | 0.001 | 0.030 | 0.029 | 0.063 | 0.086 | 0.094 |
| adjusted $R^2$ | 0.031 | 0.001 | 0.030 | 0.029 | 0.063 | 0.086 | 0.094 |

*Note:* $^*p < 0.05$; $^{**}p < 0.01$; $^{***}p < 0.005$.

**Table 2.** Associations with average wages. Linear regressions for (log) average wages at the city–industry level as a function of city (log) population size, (log) of average years of schooling, (log) of the inherent industry complexity, (log) of the collective know-how of city, controlling for economic and product complexity indices. Regression table shows only the year 2016.

| | dependent variable: log(ave. wage) | | | | | | |
|---|---|---|---|---|---|---|---|
| | (1) | (2) | (3) | (4) | (5) | (6) | (7) |
| log(city population) | 0.105*** | | | | | 0.088*** | 0.071*** |
| | t = 34.241 | | | | | t = 27.370 | t = 18.535 |
| log(ave. yrs. schooling) | | 0.812*** | | | 0.547*** | 0.371*** | 0.231*** |
| | | t = 18.700 | | | t = 11.473 | t = 7.852 | t = 4.650 |
| log(industry complexity) | | | 0.359*** | | 0.341*** | 0.304*** | 0.043* |
| | | | t = 20.188 | | t = 19.381 | t = 17.583 | t = 2.389 |
| log(city collective know-how) | | | | 1.508*** | 0.995*** | 0.460*** | 0.294*** |
| | | | | t = 18.769 | t = 11.280 | t = 5.183 | t = 3.174 |
| PCI | | | | | | | 0.037*** |
| | | | | | | | t = 36.753 |
| ECI | | | | | | | 0.026*** |
| | | | | | | | t = 5.028 |
| constant | 9.243*** | 9.433*** | 10.581*** | 10.587*** | 9.804*** | 8.927*** | 9.317*** |
| | t = 234.625 | t = 152.728 | t = 2, 986.174 | t = 2, 993.696 | t = 144.627 | t = 120.891 | t = 102.799 |
| observations | 19 490 | 19 490 | 19 490 | 19 490 | 19 490 | 19 490 | 19 490 |
| $R^2$ | 0.057 | 0.018 | 0.020 | 0.018 | 0.043 | 0.078 | 0.140 |
| adjusted $R^2$ | 0.057 | 0.018 | 0.020 | 0.018 | 0.043 | 0.078 | 0.140 |

Note: *$p < 0.05$; **$p < 0.01$; ***$p < 0.005$

and becomes negatively associated, indicating that a 1% increase in the average years of schooling of individuals in a city is associated with a 0.6–0.9% *reduction* of establishment size. Such result, although somewhat unintuitive, is consistent with our theory. However, we remind the reader that average years of schooling is not a direct estimate of individual know-how, since we did not use our methodology to estimate that parameter of our model, and serves here a statistical role to reduce the effect of omitted variables.

The association between firm size and the inherent complexity of industries, however, is unexpected. We find that a 1% increase in the number of 'ingredients' of an industry is associated with approximately a 0.8% reduction in the size of establishments. Note that this negative relationship also holds for the PCI. At face value, these results may be indications of an inconsistency in our model, or an indication that there may be unaccounted sources of error or omitted variables. Together with figure 5*b*, these results suggest that investigating the determinants of establishment size across different industries, and their connection with economic complexity, is an interesting direction for future work.

Finally, the relationship with collective know-how is as expected: establishments located in cities with high levels of collective know-how are larger in size. This positive relationship is maintained even after including the ECI.

In table 2, we observe very similar patterns in wages. The coefficient for population size is again relatively stable across specifications, with a 1% increase in population size associated with a 0.07–0.10% increase in average wages in the city–industry cell. Average years of schooling is positively associated with wages, with a 1% increase leading to a 0.2–0.8% increase in wages. Also consistent with our expectations is that a 1% change in the inherent complexity of an industry is associated with approximately a 0.3% positive change in wages. Note, however, that this positive association loses some statistical significance if we control for the PCI and ECI. Finally, there is a positive relationship between average wages and collective know-how, maintained even after including the ECI. However, this relationship also loses some statistical significance after we control for city population size.

The results shown in tables 1 and 2 are unchanged across the years (see electronic supplementary material, L and figure 14). Interestingly, the time evolution of these coefficients reveals that wages and industry complexity have become more tightly associated over the years.

# 6. Discussion and concluding remarks

Work in economic complexity and evolutionary economics has proposed that economic development is the process of accumulating an ever-increasing variety of capabilities [2,6,12,56,93], as opposed to a process of increasing the intensity of a few factors of production. However, since productive capabilities are often embedded in people as tacit knowledge [49–51,94], and are therefore not always observable or identifiable, Hidalgo & Hausmann [2] first introduced a set of indicators to infer the number of capabilities present in a place (and required by firms in different economic activities) by analysing directly the high-dimensional data about the collection of products that places are able to produce. They named these indicators the ECI and the PCI. Soon after, additional research followed suit and developed other algorithms to use such high-dimensional data to compute complexity metrics (e.g. [16,20,22]).

While the idea of economic performance as a matter of capability accumulation is well rooted in theories of cultural and social evolution [62], and it is consistent with the observation that differences in economic performance across time and space seem to arise because some societies are collectively more productive than others [62,95,96], the economic complexity indices proposed to this date rest on less solid ground, and their use remains therefore contested [33,34,97]. So far, there is no theory that demonstrates that the mathematical manipulations that underlie these algorithms actually quantify anything like 'number of capabilities'. Schetter [31] has shown that ECI (or a measure closely related to it) may rank places according to some underlying productivity (conditional on some assumptions about the structure of the data), but others have shown that ECI is better interpreted as an index to quantify how much a place is specialized on a few economic activities [21,98,99]. If we believe the economic performance of places is related to the number of capabilities present in them, then we must devise reliable ways of estimating them. As indices of economic complexity become more commonly used in the policy sphere [100–102], we must ensure to have sensible theories, robust methodologies, and reliable interpretations of metrics that actually quantify capabilities. In this paper, we have proposed a mathematical model combined with a data-driven approach to address these issues.

Our contributions consist of three main results. First, using the economic complexity framework, we derived a probabilistic model that explains the patterns of industrial employment in cities as a process of capability recombination. Second, based on the model, we proposed a simple statistical method to estimate the economic complexity variables that emerge from this model, such as the complexity of industries and the collective know-how of cities. And third, using estimates of industry complexity and city know-how, we showed that these variables have strong statistical associations with measures of economic performance such as average wages and average establishment sizes.

In the model, individual know-how, industry complexity and urban collective know-how interact as a product in an exponential function, which makes them tightly intertwined. We showed how, from this mathematical interaction, the triangular pattern in the matrix of places and economic activities, which has been widely observed and investigated in the literature on economic complexity (e.g. [2,21,37,56,103]) emerges (figure 2). We also showed how this interaction may be interpreted as a complementarity that would tend to sort complex industries and skilled individuals together into diverse cities. This is a topic that has been widely investigated in the urban economics literature (e.g. [70,76]) where it has been noted that this sorting occurs as a result of the better possibilities of matching employers with employees in large cities (e.g. [78]), and thus it is not surprising to find such effect in our model as well. While further exploration is needed to establish connections with other models of cities, we nevertheless presented supporting evidence that the functional form predicted by the model is statistically superior to some alternative models. The main lesson from our model is that a simple probabilistic 'production-recipes' approach of recombination of inputs in cities, with little assumptions about the nature of those inputs or the presence of market forces, can go a long way into providing firm footing for considering economic complexity quantities as fundamental (as opposed to ad hoc summary statistics of high-dimensional data), for establishing a method for estimating them, and for providing reliable interpretations of these quantities as meaningful economic indicators.

A more practical aspect of our results was the method to estimate the parameters of the model from data, which can be extended to apply when individual data are available. In particular, we showed how, conditional on some manipulations of the data, complexity metrics can be estimated as fixed effects in a regression model. This method is likely to work best with social security data that track individuals and their job trajectories across places and industries. Here, however, we focused instead on analysing estimates of industry complexity and collective know-how.

We have shown that our estimates of the know-how of cities correlate with intuitive measures of urban complexity like population size and industrial diversity, and they also strongly correlate with measures of urban output, like average wages and firm sizes. Crucially, however, our estimates resulted in rankings of cities by collective know-how that were somewhat surprising, in the sense that large cities like New York or San Francisco MSAs did not rank particularly high. One explanation is that our estimates of urban collective know-how have a large standard error due to small sample sizes. But another interpretation is that measures like population size (or industrial diversity) alone do not take into account the specific combinations of people and technology in a city that can increase (or decrease) the potential for capability recombination. This is consistent with the findings by Balland & Rigby [19], for example, who found that high-complexity patenting tends to be geographically concentrated, yet not exclusively in the most populous cities; or by Sarkar *et al.* [104], who found that to understand urban economic performance the specific intra-city composition of economic and social activity and physical infrastructure should be taken into account, in addition to just population size. This suggests that our estimate of city collective know-how may capture such intra-city composition.

Building on the previous point, our model differs from previous theories in the urban scaling literature regarding the prevalence of phenomena in cities (e.g. levels of employment in industries, number of homicides in a city in a year, cases of infectious diseases) [74,105–107], which are anchored only in the suitability of the city to foster the occurrence of a given phenomenon. In previous models, the suitability of a city is determined by the networks of interactions within the city. Thus, these previous approaches neglected the idea that different activities (more precisely, the individuals engaged in them) would respond differently to the same urban context. The broad argument from these previous theories stated that since interactions in a network scale faster than linearly with the number of agents, urban indicators that are the result of social interactions would also scale superlinearly with population size. As a consequence, the canonical urban scaling model [74] predicts that all measures of output *per capita* will scale with the square root of city population density, regardless of any consideration about spatial equilibrium or budget constraints (see electronic supplementary material, G). Our model, on the other hand, adds a mechanism in which only the right combination of factors leads to the creation of employment, and since industries differ in the

factors required, they will scale differently. Instead of emphasizing that interactions scale faster than the number of individuals in a network, our model emphasizes the diversity of types of interactions growing with the number of interactions. The collective know-how, $r$, quantifies what really matters about the context in which individuals are embedded: that some contexts are more complementary to some individuals and to some activities than others.

Our results have two main limitations, one theoretical and one empirical. First, our model does not specify how places acquire capabilities, or how capabilities emerge or evolve. Hence, we currently lack a clear specification for how our model predicts economic change. That said, preliminary analysis suggests that our complexity variables are promising in explaining economic growth in cities (see electronic supplementary material, M for an analysis of how our estimate of collective know-how correlates with GDP *per capita* level and growth). Incorporating time into our model is part of future work, and one crucial insight can be drawn from the regression analyses we have presented here linking our model with the larger literature on economic complexity: we found that the *number* of capabilities (present in cities or required by industries) are as important as the *type* of capabilities. This result was found by comparing the regression coefficients of ECI and PCI proposed in [2] against our 'drivers' (column 7 in tables 1 and 2, respectively). As is observed in those regressions, the inclusion of the ECI and PCI is statistically orthogonal to our measures of collective know-how and industry complexity. Assuming our model is correct, this finding is inconsistent with the conventional interpretation of the ECI and PCI. However, the fact that they are still statistically significant provides evidence that these quantities are capturing different information about economic activities in places. Namely, our estimates measure information about the number of capabilities in places and industries, while the PCI and ECI capture the specialization patterns about which types of capabilities are present in a place, or required by an industry (consistent with what is argued by [21]). Hence, the important conclusion here is that both the number and the type of capabilities affect economic performance. The generalization of the model must thus require an explicit inclusion of technological similarities between industries as has been studied in the literature of related variety and related diversification [11,12,94,108–110], and presumably include migration dynamics between cities to model the flow of capabilities (e.g. [52,111–113]). As for the empirical limitation, we must highlight that we have tested the validity of our model using data from US metropolitan areas. While the US urban system has been predominantly the focus of study for understanding cities, there is evidence that the regularities found in it are not necessarily universal (e.g. [114,115]). Hence, applying this model and the method of estimating economic complexity metrics to explain urban outcomes in other parts of the world stands as an important avenue for future research.

We believe our work contributes to the literature on economic complexity by providing conceptual and theoretical improvements that lead to a deeper understanding of cities as heterogeneous places where diverse individuals engage in an interconnected network of complex phenomena.

Data accessibility. Electronic supplementary material, D, 'Data sources', explains how our research is based on publicly available data, and details all the data sources used and how to access them.

Authors' contributions. A.G.-L. carried out the mathematical derivations, data curation, statistical analysis, writing original draft, writing final draft, editing, conceptualization and design of study. O.P.-L. contributed with the writing of the original draft, writing of the final draft, editing, conceptualization and design of study.

Competing interests. We declare we have no competing interests.

Funding. No funding has been received for this article.

Acknowledgements. We thank F. Neffke, R. Hausmann, D. Diodato, C. Bottai, U. Schetter and participants at the Growth Laboratory seminar at the Center for International Development, for helpful discussions and comments.

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
