## [Peer Review File · Royal Society Open Science]

Review History

RSOS-210670.R0 (Original submission)

Review form: Reviewer 1

Is the manuscript scientifically sound in its present form?

Yes

Are the interpretations and conclusions justified by the results?

Yes

Is the language acceptable?

Yes

Do you have any ethical concerns with this paper?

No

Have you any concerns about statistical analyses in this paper?

No

Recommendation?

Major revision is needed (please make suggestions in comments)

Comments to the Author(s)

Review report for “Estimating the drivers of urban economic complexity and their connection to economic performance” by Gomez-Lievano and Patterson-Lomba.

This paper estimates the capabilities that drive urban economic development and explores their association with economic development. Specifically, the authors proposed a model to illustrate how to infer three important quantities, namely, the complexity of urban activities, the knowhow of individual workers, and the city-wide collective knowhow, from the probability that an individual in a city is employed in a given urban economic activity. Using US employment data at the MSA level, the authors showed that the theory-informed probability functions can explain some results on urban scaling and economic complexity.

Understanding the drivers of economic complexity across scales is an interesting issue, which has important implications in terms of urban and regional development. In my view, the authors presented a very nice work by developing the model and by showing some supporting empirical evidence. This paper for sure adds to the literature, and I advocate it for final publication. At the same time, I would like to provide some comments and suggestions, which may be helpful for the authors to further strengthen their work.

1. In the introduction section, the authors presented broad conservation along with the patterns of urban economic development and economic complexity indices. Besides papers about US regions, which is aligned with the employment data used in this paper, there are also some other relevant works at different scales, from nations to regions to cities, such as Japan [Chakraborty, A., Inoue, H., & Fujiwara, Y. (2020). Economic complexity of prefectures in Japan. *PloS one*, 15(8), e0238017], and China [Gao, J., & Zhou, T. (2018). Quantifying China’s regional economic complexity. *Physica A*, 492, 1591-1603]. Many other works are also well summarized in the recent review paper [Hidalgo, C. A. (2021). Economic complexity theory and applications. *Nature Reviews Physics*, 3, 92-113].
2. Specifically about the economic complex index, besides the review paper, some recent research papers are also notable, such as [Mealy, P., Farmer, J. D., & Teytelboym, A. (2019). Interpreting economic complexity. *Science advances*, 5(1), eaau1705], [Tacchella, A., Mazzilli, D., & Pietronero, L. (2018). A dynamical systems approach to gross domestic product forecasting. *Nature Physics*, 14(8), 861-865], and [Sciarra, C., Chiarotti, G., Ridolfi, L., & Laio, F. (2020). Reconciling contrasting views on economic complexity. *Nature communications*, 11(1), 1-10]. Some of these works on economic complexity and urban scaling laws the authors covered (page 2, lines 30-35) are also briefly summarized in a review paper [Gao, J., Zhang, Y. C., & Zhou, T. (2019). Computational socioeconomics. *Physics Reports*, 817, 1-104]. Maybe these literatures are also helpful for improving the authors’ overall narratives in the introduction section.
3. I found the structure of the introduction section a bit different from traditional ones, as the authors first presented their three findings and then introduced previous works on urban scaling and the theory of economic complexity (page 2). My feeling is that it may be better to flip the structure, which will better navigate readers through the literature, the motivation of this paper, and the core contributions. Moreover, in the top of page 2, when the authors introduced the three findings, there are many new concepts to more general audiences. It may be better to do a brief lecture and elaborate more on the differences between knowledge, knowhow, know-what, know-why, know-who, etc [Coscia, M., Neffke, F. M., & Hausmann, R. (2020). Knowledge diffusion in the network of international business travel. *Nature Human Behaviour*, 4(10), 1011-1020]. These concepts may not be apparent to most readers.

4. On page 2, the authors posited the existence of three underlying variables of economic complexity. I am wondering how these concepts (the intrinsic "complexity" of the industry, a measure of "individual knowhow", and a measure of "collective knowhow") are related to previous literatures. It seems that "individual knowhow" is related to occupation-specific and industry-specific knowledge, and "collective knowhow" is related to location-specific knowledge [see, for example, Jara-Figueroa, C., Jun, B., Glaeser, E. L., & Hidalgo, C. A. (2018). The role of industry-specific, occupation-specific, and location-specific knowledge in the growth and survival of new firms. *Proceedings of the National Academy of Sciences*, 115(50), 12646-12653]. It may be helpful to clarify these differences as well as commonalities.

5. On page 2, Line 83, the authors presented that "The complexity M_f of an industry, however, should not necessarily be associated with the average knowhow of the workers employed in f ." The authors also presented an example. Yet, it remains unclear to me for getting the point. It would be appreciated if the authors are willing to help further elaborate. The same applies to page 4 Line 89, "Although s_i can be interpreted as the level of schooling or education, it is meant to capture the breadth rather than the depth of knowledge."

6. For all figures, the font sizes of both x-ticks and y-ticks are too small to read, for example, in Figure 1 and Figure 2. The authors are suggested to largely increase the font size of all ticks. In addition, it may be better to label the two panels of Figure 1 with (A) and (B) rather than referencing them using left and right.

7. In Figure 2, it may be better to add the correlation value and R^2 to the main panels (B) and (C). The same applies to Figure 3 and Figure 4. In the inset of panel (C), there seems to be an outlier around 2001. Could the authors explain why? In Figure 3, the variable in the title of y-axis starts with (-1). I am wondering if there is a better way to present this or it should go in this way.

8. In Table 1 and Table 2, the values of R^2 appear to be relatively small, ranging from 0.001 to 0.140. Considering that the regression coefficients of independent variables hold very significant, I am wondering if this is because of the relatively large sample size. Could the authors help understand why R^2 is that small, and what this tells us?

9. In the reference section, the authors are suggested to go through all references and make corrections. For example, to use the same name for PNAS, to add page number for papers published in Scientific Reports and PLOS ONE (and Ref [7] and Ref [20]), to capitalize journal names, to capitalize country names like the US in Ref [8], to lowercase paper titles, and to include the full page ranges (Ref [38]).

10. One of the authors is affiliated with a private firm. Under the conflict of interest, the authors presented "I/We declare we have no competing interests". It may be necessary to make sure the statement is suitable.

11. Very minor issues:

- (a) Page 2, Line 23, a first-principles model -> a first-principal model.
- (b) Page 3, Line 75, there may be a period after "however".

Review form: Reviewer 2

Is the manuscript scientifically sound in its present form?

Yes

Are the interpretations and conclusions justified by the results?

Yes

Is the language acceptable?

Yes

Do you have any ethical concerns with this paper?

No

Have you any concerns about statistical analyses in this paper?

Yes

Recommendation?

Major revision is needed (please make suggestions in comments)

Comments to the Author(s)

Review: Estimating the drivers of urban economic complexity and their connection to economic performance RSOS-210670

The paper seeks to build measures for the three quantities of complexity of urban activities, knowhow of individuals, and city-wide collective knowhow and links these to metrics of places' economic performances. The paper is well-written and convinces with comprehensive and elaborated empirical modelling. Yet, the paper is strongly embedded into the literature on urban scaling and cities as complex systems. This makes its contribution limited to this literature and it does not offer many insights for non-specialists and scholars outside this literature stream. Consequently, I am not sure how well it fits the interest of readers of the Royal Society Open Science. Conditional on this, I recommend major revisions addressing the points below.

1. My major concern relates to the overall message of the paper. From the logic of the urban scaling literature, I do understand the purpose of the paper. However, from the perspectives of scholars interested in urban development and spatial processes, it is not clear what the paper really offers. Most importantly, the proposed model does not explain the growth of cities nor their change or development over time. It does not establish causal relations (or tests for these). From my reading, it presents a number of metrics that correlate with certain features of cities. None of the correlations is surprising or gives insights into aspects that have not been explored in more detailed analyses. Consequently, I strongly urge the authors to make clear what the paper has to offer that is of interest to general readers outsider their particular literature.
2. In particular in the discussion section, the authors engage in causal interpretations of the relation between their measures and other metrics. For instance, on page 16, line 356, the authors claim that they have shown how "this interaction ... explained WHY a triangular pattern and WHY complex industries and skilled individuals would tend to concentrate in diverse cities". That is incorrect. The authors do not present a locational choice model for individuals or firms, nor do they explain the emergence of skills or firms. Consequently, they cannot answer the "Why". Their approach presents a description of the outcome of causal mechanisms without giving insights into these. The authors are advised to reassess their text and stay away from such causal claims or engage in true causal modelling and econometrics that identify the direction of the relations.
3. To offer some connections to the well-established literatures on urban growth, the authors are advised to make references to the contemporary literature on "location choice theories", "urban growth", and "regional diversification", all of which have a lot to offer for enriching the paper. In particular the works of location choice and agglomeration economies will provide clearer insights into what locations attract an creates what kind of talent and vice versa (Glaeser, Kallal, Scheinkam, & Shleifer, 1992; Henderson, Kuncoro, & Turner, 1995; Isard, 1956).

Similarly, the research strand on related diversification gives detailed insights into how different forms of diversity shape economic structures in a path dependent manner (Boschma, 2017; Boschma & Frenken, 2011). In addition, I recommend the authors to evaluate their work against the traditional ideas of “spatial sorting” that explain processes related to the matching of industries and (skilled) individuals (Mion & Naticchioni, 2009). Given all these works, it is hard to accept that the authors write on page 6: “Presently, we lack a detailed theory about the dynamical laws of these drivers and how they relate to one another”.

4. The authors postulate that someone uses population size “naively” as “proxy for urban knowledge”. Outside the literature on urban scaling, no one is doing that, as Economic Geographers and Regional Scientists are very much aware of the heterogeneity of urban areas across space and time. I suggest the authors to pay attention to this and link their work to these well-established ideas.

5. On page 16, line 340-353, the authors present what the paper contributes. However, they do not discuss what are the benefits of these contributions, i.e., in what sense are they better or more advantageous than what has been previously known?

6. In the end, the authors write that the major conclusion of the paper is that “both the number and the type of capabilities affect economic performance. The generalization of the model must thus require an explicit inclusion of technological similarities between industries ...” (p. 18, line 416). This is very much what the literature on related variety argues for a long time (Boschma & Frenken, 2011; Frenken & Boschma, 2007; Frenken, Van Oort, & Verburg, 2007). I am surprised to see no reference to this literature and discussion on what this implies for the future development of the proposed approach.

7. There are some claims in the paper that are hard to accept and that just seem to be made to make the model fit. For instance, on page 9, line 230, it is claimed that “more years of educational attainment are usually a measure of specialization, but they are also an indication of a person’s competences to perform several tasks”. Besides the fact that both parts of the sentence somewhat contradict each other, I also don’t believe the second part. The authors need to add some empirical support or references to support their claim. Similar is true for the claim on page 5, line 127, that diverse cities are “disproportionately more competitive in complex economic activities”. Please add support for this or remove.

8. Moreover, it is recommend to check the rest of the paper and remove or back-up such claims.

9. Please point out that the empirical results are restricted to the US, which is known to be non-representative for most countries in the world.

10. In the beginning of the work, e.g., page 3, the paper reads very much like “complexity” being more or less the same as “diversity”. This is of course not correct and the authors need to revise this part to avoid this impression.

11. With respect to the empirics, I recommend to at least include tests for spatial autocorrelation and cluster standard errors at the MSA level, to account for the fact that the observations are not independent of each other.

Decision letter (RSOS-210670.R0)

Dear Dr Gomez-Lievano

The Editors assigned to your paper RSOS-210670 "Estimating the drivers of urban economic complexity and their connection to economic performance" have now received comments from

reviewers and would like you to revise the paper in accordance with the reviewer comments and any comments from the Editors. Please note this decision does not guarantee eventual acceptance.

Please submit your revised manuscript and required files (see below) no later than 21 days from today's (ie 23-Jun-2021) date. Note: the ScholarOne system will 'lock' if submission of the revision is attempted 21 or more days after the deadline. If you do not think you will be able to meet this deadline please contact the editorial office immediately.

Kind regards,
Royal Society Open Science Editorial Office
Royal Society Open Science
openseience@royalsociety.org

on behalf of Professor Tim Rogers (Associate Editor) and Nick Pearce (Subject Editor)
openseience@royalsociety.org

Reviewer comments to Author:

Reviewer: 1

Comments to the Author(s)

Review report for "Estimating the drivers of urban economic complexity and their connection to economic performance" by Gomez-Lievano and Patterson-Lomba.

This paper estimates the capabilities that drive urban economic development and explores their association with economic development. Specifically, the authors proposed a model to illustrate how to infer three important quantities, namely, the complexity of urban activities, the knowhow of individual workers, and the city-wide collective knowhow, from the probability that an individual in a city is employed in a given urban economic activity. Using US employment data at the MSA level, the authors showed that the theory-informed probability functions can explain some results on urban scaling and economic complexity.

Understanding the drivers of economic complexity across scales is an interesting issue, which has important implications in terms of urban and regional development. In my view, the authors

presented a very nice work by developing the model and by showing some supporting empirical evidence. This paper for sure adds to the literature, and I advocate it for final publication. At the same time, I would like to provide some comments and suggestions, which may be helpful for the authors to further strengthen their work.

1. In the introduction section, the authors presented broad conservation along with the patterns of urban economic development and economic complexity indices. Besides papers about US regions, which is aligned with the employment data used in this paper, there are also some other relevant works at different scales, from nations to regions to cities, such as Japan [Chakraborty, A., Inoue, H., & Fujiwara, Y. (2020). Economic complexity of prefectures in Japan. *PloS one*, 15(8), e0238017], and China [Gao, J., & Zhou, T. (2018). Quantifying China's regional economic complexity. *Physica A*, 492, 1591-1603]. Many other works are also well summarized in the recent review paper [Hidalgo, C. A. (2021). Economic complexity theory and applications. *Nature Reviews Physics*, 3, 92-113].
2. Specifically about the economic complex index, besides the review paper, some recent research papers are also notable, such as [Mealy, P., Farmer, J. D., & Teytelboym, A. (2019). Interpreting economic complexity. *Science advances*, 5(1), eaau1705], [Tachella, A., Mazzilli, D., & Pietronero, L. (2018). A dynamical systems approach to gross domestic product forecasting. *Nature Physics*, 14(8), 861-865], and [Sciarra, C., Chiarotti, G., Ridolfi, L., & Laio, F. (2020). Reconciling contrasting views on economic complexity. *Nature communications*, 11(1), 1-10]. Some of these works on economic complexity and urban scaling laws the authors covered (page 2, lines 30-35) are also briefly summarized in a review paper [Gao, J., Zhang, Y. C., & Zhou, T. (2019). Computational socioeconomics. *Physics Reports*, 817, 1-104]. Maybe these literatures are also helpful for improving the authors' overall narratives in the introduction section.
3. I found the structure of the introduction section a bit different from traditional ones, as the authors first presented their three findings and then introduced previous works on urban scaling and the theory of economic complexity (page 2). My feeling is that it may be better to flip the structure, which will better navigate readers through the literature, the motivation of this paper, and the core contributions. Moreover, in the top of page 2, when the authors introduced the three findings, there are many new concepts to more general audiences. It may be better to do a brief lecture and elaborate more on the differences between knowledge, knowhow, know-what, know-why, know-who, etc [Coscia, M., Neffke, F. M., & Hausmann, R. (2020). Knowledge diffusion in the network of international business travel. *Nature Human Behaviour*, 4(10), 1011-1020]. These concepts may not be apparent to most readers.
4. On page 2, the authors posited the existence of three underlying variables of economic complexity. I am wondering how these concepts (the intrinsic "complexity" of the industry, a measure of "individual knowhow", and a measure of "collective knowhow") are related to previous literatures. It seems that "individual knowhow" is related to occupation-specific and industry-specific knowledge, and "collective knowhow" is related to location-specific knowledge [see, for example, Jara-Figueroa, C., Jun, B., Glaeser, E. L., & Hidalgo, C. A. (2018). The role of industry-specific, occupation-specific, and location-specific knowledge in the growth and survival of new firms. *Proceedings of the National Academy of Sciences*, 115(50), 12646-12653]. It may be helpful to clarify these differences as well as commonalities.
5. On page 2, Line 83, the authors presented that "The complexity M_f of an industry, however, should not necessarily be associated with the average knowhow of the workers employed in f ." The authors also presented an example. Yet, it remains unclear to me for getting the point. It would be appreciated if the authors are willing to help further elaborate. The same applies to

page 4 Line 89, “Although si can be interpreted as the level of schooling or education, it is meant to capture the breadth rather than the depth of knowledge.”

6. For all figures, the font sizes of both x-ticks and y-ticks are too small to read, for example, in Figure 1 and Figure 2. The authors are suggested to largely increase the font size of all ticks. In addition, it may be better to label the two panels of Figure 1 with (A) and (B) rather than referencing them using left and right.

7. In Figure 2, it may be better to add the correlation value and R^2 to the main panels (B) and (C). The same applies to Figure 3 and Figure 4. In the inset of panel (C), there seems to be an outlier around 2001. Could the authors explain why? In Figure 3, the variable in the title of y-axis starts with (-1). I am wondering if there is a better way to present this or it should go in this way.

8. In Table 1 and Table 2, the values of R^2 appear to be relatively small, ranging from 0.001 to 0.140. Considering that the regression coefficients of independent variables hold very significant, I am wondering if this is because of the relatively large sample size. Could the authors help understand why R^2 is that small, and what this tells us?

9. In the reference section, the authors are suggested to go through all references and make corrections. For example, to use the same name for PNAS, to add page number for papers published in Scientific Reports and PLOS ONE (and Ref [7] and Ref [20]), to capitalize journal names, to capitalize country names like the US in Ref [8], to lowercase paper titles, and to include the full page ranges (Ref [38]).

10. One of the authors is affiliated with a private firm. Under the conflict of interest, the authors presented “I/We declare we have no competing interests”. It may be necessary to make sure the statement is suitable.

11. Very minor issues:

(a) Page 2, Line 23, a first-principles model -> a first-principal model.

(b) Page 3, Line 75, there may be a period after “however”.

Reviewer: 2

Comments to the Author(s)

Review: Estimating the drivers of urban economic complexity and their connection to economic performance RSOS-210670

The paper seeks to build measures for the three quantities of complexity of urban activities, knowhow of individuals, and city-wide collective knowhow and links these to metrics of places' economic performances. The paper is well-written and convinces with comprehensive and elaborated empirical modelling. Yet, the paper is strongly embedded into the literature on urban scaling and cities as complex systems. This makes its contribution limited to this literature and it does not offer many insights for non-specialists and scholars outside this literature stream. Consequently, I am not sure how well it fits the interest of readers of the Royal Society Open Science. Conditional on this, I recommend major revisions addressing the points below.

1. My major concern relates to the overall message of the paper. From the logic of the urban scaling literature, I do understand the purpose of the paper. However, from the perspectives of scholars interested in urban development and spatial processes, it is not clear what the paper really offers. Most importantly, the proposed model does not explain the growth of cities nor their change or development over time. It does not establish causal relations (or tests for these). From my reading, it presents a number of metrics that correlate with certain features of cities.

None of the correlations is surprising or gives insights into aspects that have not been explored in more detailed analyses. Consequently, I strongly urge the authors to make clear what the paper has to offer that is of interest to general readers outsider their particular literature.

2. In particular in the discussion section, the authors engage in causal interpretations of the relation between their measures and other metrics. For instance, on page 16, line 356, the authors claim that they have shown how “this interaction ... explained WHY a triangular pattern and WHY complex industries and skilled individuals would tend to concentrate in diverse cities”.

That is incorrect. The authors do not present a locational choice model for individuals or firms, nor do they explain the emergence of skills or firms. Consequently, they cannot answer the “Why”. Their approach presents a description of the outcome of causal mechanisms without giving insights into these. The authors are advised to reassess their text and stay away from such causal claims or engage in true causal modelling and econometrics that identify the direction of the relations.

3. To offer some connections to the well-established literatures on urban growth, the authors are advised to make references to the contemporary literature on “location choice theories”, “urban growth”, and “regional diversification”, all of which have a lot to offer for enriching the paper. In particular the works of location choice and agglomeration economies will provide clearer insights into what locations attract an creates what kind of talent and vice versa (Glaeser, Kallal, Scheinkam, & Shleifer, 1992; Henderson, Kuncoro, & Turner, 1995; Isard, 1956). Similarly, the research strand on related diversification gives detailed insights into how different forms of diversity shape economic structures in a path dependent manner (Boschma, 2017; Boschma & Frenken, 2011). In addition, I recommend the authors to evaluate their work against the traditional ideas of “spatial sorting” that explain processes related to the matching of industries and (skilled) individuals (Mion & Naticchioni, 2009). Given all these works, it is hard to accept that the authors write on page 6: “Presently, we lack a detailed theory about the dynamical laws of these drivers and how they relate to one another”.

4. The authors postulate that someone uses population size “naively” as “proxy for urban knowledge”. Outside the literature on urban scaling, no one is doing that, as Economic Geographers and Regional Scientists are very much aware of the heterogeneity of urban areas across space and time. I suggest the authors to pay attention to this and link their work to these well-established ideas.

5. On page 16, line 340-353, the authors present what the paper contributes. However, they do not discuss what are the benefits of these contributions, i.e., in what sense are they better or more advantageous than what has been previously known?

6. In the end, the authors write that the major conclusion of the paper is that “both the number and the type of capabilities affect economic performance. The generalization of the model must thus require an explicit inclusion of technological similarities between industries ...” (p. 18, line 416). This is very much what the literature on related variety argues for a long time (Boschma & Frenken, 2011; Frenken & Boschma, 2007; Frenken, Van Oort, & Verburg, 2007). I am surprised to see no reference to this literature and discussion on what this implies for the future development of the proposed approach.

7. There are some claims in the paper that are hard to accept and that just seem to be made to make the model fit. For instance, on page 9, line 230, it is claimed that “more years of educational attainment are usually a measure of specialization, but they are also an indication of a person’s competences to perform several tasks”. Besides the fact that both parts of the sentence somewhat contradict each other, I also don’t believe the second part. The authors need to add some empirical support or references to support their claim. Similar is true for the claim on page 5, line 127, that diverse cities are “disproportionately more competitive in complex economic activities”. Please add support for this or remove.

8. Moreover, it is recommend to check the rest of the paper and remove or back-up such claims.

9. Please point out that the empirical results are restricted to the US, which is known to be non-representative for most countries in the world.

10. In the beginning of the work, e.g., page 3, the paper reads very much like “complexity” being more or less the same as “diversity”. This is of course not correct and the authors need to revise this part to avoid this impression.

11. With respect to the empirics, I recommend to at least include tests for spatial autocorrelation and cluster standard errors at the MSA level, to account for the fact that the observations are not independent of each other.

===PREPARING YOUR MANUSCRIPT===

===PREPARING YOUR REVISION IN SCHOLARONE===

Author's Response to Decision Letter for (RSOS-210670.R0)

See Appendix A.

RSOS-210670.R1 (Revision)

Review form: Reviewer 1

Is the manuscript scientifically sound in its present form?

Yes

Are the interpretations and conclusions justified by the results?

Yes

Is the language acceptable?

Yes

Do you have any ethical concerns with this paper?

No

Have you any concerns about statistical analyses in this paper?

No

Recommendation?

Accept as is

Comments to the Author(s)

Review report for "Estimating the drivers of urban economic complexity and their connection to economic performance" by Gomez-Lievano and Patterson-Lomba.

I would like to thank the authors for considering my comments and suggestions on the previous version of this paper. All my previous concerns have been well addressed by the authors in this version. In my view, this paper has been substantially improved, and the authors have clarified suitably where the paper stands and what contributions they were trying to make. Overall, this reinforces my initial feeling and recommendation that this paper adds to the literature decently and can be considered for publication in the Royal Society Open Science.

Limited by my narrow expertise, I have no further technical comments to provide in this round, and I think the paper reads well, without checking every detail due to the limit on time. During the next iteration, however, it will be also great if the authors can notice some minor issues. For example, (1) The tick sizes of Figure 6 are still too small. The authors can repeat what they did for Figure 4 and Figure 5, namely, increasing the font sizes to make them more visible. The ticks of some inset figures are also too small, including Figures 4BC, 5BC, and 6ABCD. (2) Page 8 Line 185, around section 3.4, "We refer to refer to the (negative) the exponent". I assume there is a repeated "refer to", which should be removed. (3) The page information of some references should be updated. For example, Google Scholar often returns a wrong page number for some Nature family journals including Nature Human Behavior and Nature Communications, see, for example, the current Refs [8], [16], [22], [34], and many others. These page numbers are not starting from 1. For some conference papers, the paper titles are missing, such as Ref [13]. For some very recent papers, their page numbers are now available, such as Ref [24]. All these very minor issues can be fixed at the proof stage as well.

Review form: Reviewer 2

Is the manuscript scientifically sound in its present form?

Yes

Are the interpretations and conclusions justified by the results?

Yes

Is the language acceptable?

Yes

Do you have any ethical concerns with this paper?

No

Have you any concerns about statistical analyses in this paper?

No

Recommendation?

Accept with minor revision (please list in comments)

Comments to the Author(s)

Review: RSOS-210670.R1

My responses to the authors' replies to my comments.

- The authors show now that the work is indeed fitting to RSOS.

1. My major concern relates to the overall message of the paper. From the logic of the urban scaling literature, I do understand the purpose of the paper. However, from the perspectives of scholars interested in urban development and spatial processes, it is not clear what the paper really offers. Most importantly, the proposed model does not explain the growth of cities nor their change or development over time. It does not establish causal relations (or tests for these). From my reading, it presents a number of metrics that correlate with certain features of cities. None of the correlations is surprising or gives insights into aspects that have not been explored in more detailed analyses. Consequently, I strongly urge the authors to make clear what the paper has to offer that is of interest to general readers outsider their particular literature.

- The authors respond in a convincing fashion to this remark. However, if possible, I really would love to see authors being able to integrate (in a slightly adapted form) this part of their reply starting with: "Still, it is worth repeating some of the message there, ..." to "In light of this, the simplicity of our static model is not a bug but a feature.", into the introduction. It really helps positioning the paper. In my view, it is actually clearer that the corresponding part in the introduction itself. I like the clarification about the "growth" issue. I also appreciate the explorative regression analysis on GDP growth as well as the better structuring using subheadings.

2. In particular in the discussion section, the authors engage in causal interpretations of the relation between their measures and other metrics. For instance, on page 16, line 356, the authors claim that they have shown how "this interaction ... explained WHY a triangular pattern and WHY complex industries and skilled individuals would tend to concentrate in diverse cities". That is incorrect. The authors do not present a locational choice model for individuals or firms, nor do they explain the emergence of skills or firms. Consequently, they cannot answer the "Why". Their approach presents a description of the outcome of causal mechanisms without

giving insights into these. The authors are advised to reassess their text and stay away from such causal claims or engage in true causal modelling and econometrics that identify the direction of the relations.

- Ok, now, I better understand what the authors are after. Thanks for clarifying and adapting the text.

3. To offer some connections to the well-established literatures on urban growth, the authors are advised to make references to the contemporary literature on “location choice theories”, “urban growth”, and “regional diversification”, all of which have a lot to offer for enriching the paper. In particular the works of location choice and agglomeration economies will provide clearer insights into what locations attract and creates what kind of talent and vice versa (Glaeser, Kallal, Scheinkam, & Shleifer, 1992; Henderson, Kuncoro, & Turner, 1995; Isard, 1956). Similarly, the research strand on related diversification gives detailed insights into how different forms of diversity shape economic structures in a path dependent manner (Boschma, 2017; Boschma & Frenken, 2011). In addition, I recommend the authors to evaluate their work against the traditional ideas of “spatial sorting” that explain processes related to the matching of industries and (skilled) individuals (Mion & Naticchioni, 2009). Given all these works, it is hard to accept that the authors write on page 6: “Presently, we lack a detailed theory about the dynamical laws of these drivers and how they relate to one another”.

- Ok, I agree with the revisions and the reasons for not including some of these references. Just to be transparent as well, Isard (1956) refers to this classic:
<https://www.worldcat.org/title/location-and-space-economy-a-general-theory-relating-to-industrial-location-market-areas-land-use-trade-and-urban-structure/oclc/229658>

4. The authors postulate that someone uses population size “naively” as “proxy for urban knowledge”. Outside the literature on urban scaling, no one is doing that, as Economic Geographers and Regional Scientists are very much aware of the heterogeneity of urban areas across space and time. I suggest the authors to pay attention to this and link their work to these well-established ideas.

- Ok, maybe I’ve misread this part. It is clear now.

5. On page 16, line 340-353, the authors present what the paper contributes. However, they do not discuss what are the benefits of these contributions, i.e., in what sense are they better or more advantageous than what has been previously known?

- Ok, the response is nice and helpful. Thanks for the revision.

6. In the end, the authors write that the major conclusion of the paper is that “both the number and the type of capabilities affect economic performance. The generalization of the model must thus require an explicit inclusion of technological similarities between industries ...” (p. 18, line 416). This is very much what the literature on related variety argues for a long time (Boschma & Frenken, 2011; Frenken & Boschma, 2007; Frenken, Van Oort, & Verburg, 2007). I am surprised to see no reference to this literature and discussion on what this implies for the future development of the proposed approach.

- Ok, thanks for adding the references and clarifying the point.

7. There are some claims in the paper that are hard to accept and that just seem to be made to make the model fit. For instance, on page 9, line 230, it is claimed that “more years of educational attainment are usually a measure of specialization, but they are also an indication of

a person's competences to perform several tasks". Besides the fact that both parts of the sentence somewhat contradict each other, I also don't believe the second part. The authors need to add some empirical support or references to support their claim. Similar is true for the claim on page 5, line 127, that diverse cities are "disproportionately more competitive in complex economic activities". Please add support for this or remove.

- Ok. I see the point and I accept the revisions.

8. Moreover, it is recommend to check the rest of the paper and remove or back-up such claims.

- Ok.

9. Please point out that the empirical results are restricted to the US, which is known to be non-representative for most countries in the world.

- Ok. This comment has been sufficiently addressed as well.

10. In the beginning of the work, e.g., page 3, the paper reads very much like "complexity" being more or less the same as "diversity". This is of course not correct and the authors need to revise this part to avoid this impression.

- Ok. Accepted.

11. With respect to the empirics, I recommend to at least include tests for spatial autocorrelation and cluster standard errors at the MSA level, to account for the fact that the observations are not independent of each other.

- While I have to disagree with the authors on the importance of considering spatial dependencies (if existent, the currently used ways of assessing significance are invalid), and also on the idea that a low p-value cannot be altered into levels of insignificance due to spatial dependencies; I agree that such an empirical strategy will overload the paper. Moreover, it is not central to the paper. Consequently, I consider this point as delt with as well.

Decision letter (RSOS-210670.R1)

Dear Dr Gomez-Lievano,

It is a pleasure to accept your manuscript entitled "Estimating the drivers of urban economic complexity and their connection to economic performance" in its current form for publication in Royal Society Open Science. The comments of the reviewer(s) who reviewed your manuscript are included at the foot of this letter.

You can expect to receive a proof of your article in the near future. Please contact the editorial office (openscience@royalsociety.org) and the production office (openscience_proofs@royalsociety.org) to let us know if you are likely to be away from e-mail

contact -- if you are going to be away, please nominate a co-author (if available) to manage the proofing process, and ensure they are copied into your email to the journal.

on behalf of Professor Tim Rogers (Associate Editor) and Nick Pearce (Subject Editor)
openscience@royalsociety.org

Editor comments:

The authors have substantially revised the paper in response to the peer reviews, and have paid particular and detailed attention to dealing with substantive concerns about the literatures to which the paper relates, the question of its causal claims and scope, and its appropriateness to RSOS. The authors are now much clearer about what the paper claims. I am persuaded that these responses are sufficient to render the paper acceptable for publication.

Reviewer comments to Author:

Reviewer: 2

Comments to the Author(s)

Review: RSOS-210670.R1

My responses to the authors' replies to my comments.

- The authors show now that the work is indeed fitting to RSOS.

1. My major concern relates to the overall message of the paper. From the logic of the urban scaling literature, I do understand the purpose of the paper. However, from the perspectives of scholars interested in urban development and spatial processes, it is not clear what the paper really offers. Most importantly, the proposed model does not explain the growth of cities nor their change or development over time. It does not establish causal relations (or tests for these). From my reading, it presents a number of metrics that correlate with certain features of cities. None of the correlations is surprising or gives insights into aspects that have not been explored in more detailed analyses. Consequently, I strongly urge the authors to make clear what the paper has to offer that is of interest to general readers outsider their particular literature.

- The authors respond in a convincing fashion to this remark. However, if possible, I really would love to see authors being able to integrate (in a slightly adapted form) this part of their reply starting with: "Still, it is worth repeating some of the message there, ..." to "In light of this, the simplicity of our static model is not a bug but a feature.", into the introduction. It really helps positioning the paper. In my view, it is actually clearer that the corresponding part in the introduction itself. I like the clarification about the "growth" issue. I also appreciate the explorative regression analysis on GDP growth as well as the better structuring using subheadings.

2. In particular in the discussion section, the authors engage in causal interpretations of the relation between their measures and other metrics. For instance, on page 16, line 356, the authors claim that they have shown how "this interaction ... explained WHY a triangular pattern and WHY complex industries and skilled individuals would tend to concentrate in diverse cities". That is incorrect. The authors do not present a locational choice model for individuals or firms, nor do they explain the emergence of skills or firms. Consequently, they cannot answer the "Why". Their approach presents a description of the outcome of causal mechanisms without giving insights into these. The authors are advised to reassess their text and stay away from such causal claims or engage in true causal modelling and econometrics that identify the direction of the relations.

- Ok, now, I better understand what the authors are after. Thanks for clarifying and adapting the text.

3. To offer some connections to the well-established literatures on urban growth, the authors are advised to make references to the contemporary literature on "location choice theories", "urban growth", and "regional diversification", all of which have a lot to offer for enriching the paper. In particular the works of location choice and agglomeration economies will provide clearer insights into what locations attract and create what kind of talent and vice versa (Glaeser, Kallal, Scheinkam, & Shleifer, 1992; Henderson, Kuncoro, & Turner, 1995; Isard, 1956). Similarly, the research strand on related diversification gives detailed insights into how different forms of diversity shape economic structures in a path dependent manner (Boschma, 2017; Boschma & Frenken, 2011). In addition, I recommend the authors to evaluate their work against the traditional ideas of "spatial sorting" that explain processes related to the matching of industries and (skilled) individuals (Mion & Naticchioni, 2009). Given all these works, it is hard to accept that the authors write on page 6: "Presently, we lack a detailed theory about the dynamical laws of these drivers and how they relate to one another".

- Ok, I agree with the revisions and the reasons for not including some of these references. Just to be transparent as well, Isard (1956) refers to this classic:
<https://www.worldcat.org/title/location-and-space-economy-a-general-theory-relating-to-industrial-location-market-areas-land-use-trade-and-urban-structure/oclc/229658>

4. The authors postulate that someone uses population size "naively" as "proxy for urban knowledge". Outside the literature on urban scaling, no one is doing that, as Economic Geographers and Regional Scientists are very much aware of the heterogeneity of urban areas across space and time. I suggest the authors to pay attention to this and link their work to these well-established ideas.

- Ok, maybe I've misread this part. It is clear now.

5. On page 16, line 340-353, the authors present what the paper contributes. However, they do not discuss what are the benefits of these contributions, i.e., in what sense are they better or more advantageous than what has been previously known?

- Ok, the response is nice and helpful. Thanks for the revision.

6. In the end, the authors write that the major conclusion of the paper is that “both the number and the type of capabilities affect economic performance. The generalization of the model must thus require an explicit inclusion of technological similarities between industries ...” (p. 18, line 416). This is very much what the literature on related variety argues for a long time (Boschma & Frenken, 2011; Frenken & Boschma, 2007; Frenken, Van Oort, & Verburg, 2007). I am surprised to see no reference to this literature and discussion on what this implies for the future development of the proposed approach.

- Ok, thanks for adding the references and clarifying the point.

7. There are some claims in the paper that are hard to accept and that just seem to be made to make the model fit. For instance, on page 9, line 230, it is claimed that “more years of educational attainment are usually a measure of specialization, but they are also an indication of a person’s competences to perform several tasks”. Besides the fact that both parts of the sentence somewhat contradict each other, I also don’t believe the second part. The authors need to add some empirical support or references to support their claim. Similar is true for the claim on page 5, line 127, that diverse cities are “disproportionately more competitive in complex economic activities”. Please add support for this or remove.

- Ok. I see the point and I accept the revisions.

8. Moreover, it is recommend to check the rest of the paper and remove or back-up such claims.

- Ok.

9. Please point out that the empirical results are restricted to the US, which is known to be non-representative for most countries in the world.

- Ok. This comment has been sufficiently addressed as well.

10. In the beginning of the work, e.g., page 3, the paper reads very much like “complexity” being more or less the same as “diversity”. This is of course not correct and the authors need to revise this part to avoid this impression.

- Ok. Accepted.

11. With respect to the empirics, I recommend to at least include tests for spatial autocorrelation and cluster standard errors at the MSA level, to account for the fact that the observations are not independent of each other.

- While I have to disagree with the authors on the importance of considering spatial dependencies (if existent, the currently used ways of assessing significance are invalid), and also on the idea that a low p-value cannot be altered into levels of insignificance due to spatial dependencies; I agree that such an empirical strategy will overload the paper. Moreover, it is not central to the paper. Consequently, I consider this point as delt with as well.

Reviewer: 1

Comments to the Author(s)

Review report for “Estimating the drivers of urban economic complexity and their connection to economic performance” by Gomez-Lievano and Patterson-Lomba.

I would like to thank the authors for considering my comments and suggestions on the previous version of this paper. All my previous concerns have been well addressed by the authors in this version. In my view, this paper has been substantially improved, and the authors have clarified suitably where the paper stands and what contributions they were trying to make. Overall, this reinforces my initial feeling and recommendation that this paper adds to the literature decently and can be considered for publication in the Royal Society Open Science.

Limited by my narrow expertise, I have no further technical comments to provide in this round, and I think the paper reads well, without checking every detail due to the limit on time. During the next iteration, however, it will be also great if the authors can notice some minor issues. For example, (1) The tick sizes of Figure 6 are still too small. The authors can repeat what they did for Figure 4 and Figure 5, namely, increasing the font sizes to make them more visible. The ticks of some inset figures are also too small, including Figures 4BC, 5BC, and 6ABCD. (2) Page 8 Line 185, around section 3.4, “We refer to refer to the (negative) the exponent”. I assume there is a repeated “refer to”, which should be removed. (3) The page information of some references should be updated. For example, Google Scholar often returns a wrong page number for some Nature family journals including Nature Human Behavior and Nature Communications, see, for example, the current Refs [8], [16], [22], [34], and many others. These page numbers are not starting from 1. For some conference papers, the paper titles are missing, such as Ref [13]. For some very recent papers, their page numbers are now available, such as Ref [24]. All these very minor issues can be fixed at the proof stage as well.

Appendix A

Response to editor and reviewers at “Royal Society Open Science” regarding manuscript ID RSOS-210670

Dear Royal Society Open Science Editorial Office,

We greatly appreciate the review of our manuscript ID RSOS-210670 entitled “Estimating the drivers of urban economic complexity and their connection to economic performance” We thank the Reviewers for their encouraging comments (e.g., “*interesting*”, “*important implications*”, “*very nice work*”, “*this paper for sure adds to the literature, and I advocate it for final publication*”, “*well-written*”, “*convinces with comprehensive and elaborated empirical modelling*”) and their constructive suggestions.

From our perspective, most of the comments on our manuscript focused on (i) restructuring the text to better convey the overall contributions of the paper, and (ii) adding connections to related work. These comments gave us an opportunity to further clarify the exposition in the manuscript and place our work within the larger field of researchers interested in understanding urban systems. Given the comments offered by the reviewers, the main changes to our manuscript were:

- Re-organization of how some concepts are introduced
- Re-wording many parts, trying to make the sentences readable in a more linear and concise way
- Introducing citations to several new references and expanding the connections to related literature

We are cognizant that the manuscript is already long, and therefore, some of our decisions to add new material to the original manuscript were based on a judgement call about where to elaborate our responses best. Having said this, we did respond to all comments in this response letter.

We look forward to being able to publish this piece in Royal Society Open Science.

One by one, we list below the comments to our manuscript (in quoted *italic*) and then present our responses. Text that we added to the manuscript is in red.

Sincerely,
Andres Gomez-Lievano,
Oscar Patterson-Lomba

Cambridge, MA, USA

Comments and responses

Given the comments offered by the reviewers, the main changes to our manuscript are:

- Re-organization of how some concepts are introduced
- Re-wording many parts, trying to make the sentences readable in a more linear and concise way
- Introducing citations to several new references and expanding the connections to related literature

Below are our responses to all comments one by one.

Text quoted from the Reviewers is shown in double quotes (“”) and *italics*. New text added to the original manuscript due to comments from the Reviewers is shown here in double quotes (“”) and in **red**. In the manuscript, some paragraphs have been highlighted in **blue** to show parts that have been moved from one part to another.

Reviewer # 1:

1. *“In the introduction section, the authors presented broad conservation [sic] along with the patterns of urban economic development and economic complexity indices. Besides papers about US regions, which is aligned with the employment data used in this paper, there are also some other relevant works at different scales, from nations to regions to cities, such as Japan [Chakraborty, A., Inoue, H., & Fujiwara, Y. (2020). Economic complexity of prefectures in Japan. PLoS one, 15(8), e0238017], and China [Gao, J., & Zhou, T. (2018). Quantifying China’s regional economic complexity. Physica A, 492, 1591-1603]. Many other works are also well summarized in the recent review paper [Hidalgo, C. A. (2021). Economic complexity theory and applications. Nature Reviews Physics, 3, 92-113].”*

We thank the reviewer for point out these sources (the last reference was already cited in our manuscript originally as [7]). We have read them all and they are indeed relevant citations to show that complexity metrics have been successfully applied in other countries besides the US like Japan and China. These references induced us to cite additional studies in India, Mexico, and Colombia. We have added all these references (numbered as 25, 26, 27, 28, and 29) to the manuscript:

[25] A. Chakraborty, H. Inoue, Y. Fujiwara, Economic complexity of prefectures in Japan, PLoS ONE 15 (2020) e0238017.

[26] J. Gao, T. Zhou, Quantifying China’s regional economic complexity, Physica A: Statistical Mechanics and its Applications 492 (2018) 1591–1603.

[27] A. Sahasranaman, H. Jensen, Economic complexity and capabilities of Indian States, Available at SSRN 3578242 (2020).

[28] R. Hausmann, C. Pietrobelli, M. A. Santos, Place-specific determinants of income gaps: New sub-national evidence from Mexico, Journal of Business Research 131 (2021) 782–792.

2. *“Specifically about the economic complex [sic] index, besides the review paper, some recent research papers are also notable, such as [Mealy, P., Farmer, J. D., & Teytelboym, A. (2019). Interpreting economic complexity. Science advances, 5(1), eaau1705], [Tacchella, A., Mazzilli, D., & Pietronero, L. (2018). A dynamical systems approach to gross domestic product forecasting. Nature Physics, 14(8), 861-865], and [Sciarrà, C., Chiarotti, G., Ridolfi, L., & Laio, F. (2020). Reconciling contrasting views on economic complexity. Nature communications, 11(1), 1-10]. Some of these works on economic complexity and urban scaling laws the authors covered (page 2, lines 30-35) are also briefly summarized in a review paper [Gao, J.,*

Zhang, Y. C., & Zhou, T. (2019). *Computational socioeconomics. Physics Reports, 817, 1-104*. Maybe these literatures are also helpful for improving the authors' overall narratives in the introduction section."

We greatly appreciate the reviewer's comment, as this has helped us re-organize the introduction of our paper. Thanks to this comment (together with next comment and other comments from Reviewer #2), we have completely re-written the introduction. The new references suggested have been added as citations (numbered as 22, 23, 30) since they provide useful information to readers about the broad field of research in computational socioeconomics and economic complexity:

[22] C. Sciarra, G. Chiarotti, L. Ridolfi, F. Laio, Reconciling contrasting views on economic complexity, *Nature communications* 11 (2020) 1–10.

[23] A. Tacchella, D. Mazzilli, L. Pietronero, A dynamical systems approach to gross domestic product forecasting, *Nature Physics* 14 (2018) 861–865.

[30] J. Gao, Y.-C. Zhang, T. Zhou, *Computational socioeconomics, Physics Reports* 817 (2019) 1–104.

3. *"I found the structure of the introduction section a bit different from traditional ones, as the authors first presented their three findings and then introduced previous works on urban scaling and the theory of economic complexity (page 2). My feeling is that it may be better to flip the structure, which will better navigate readers through the literature, the motivation of this paper, and the core contributions. Moreover, in the top of page 2, when the authors introduced the three findings, there are many new concepts to more general audiences. It may be better to do a brief lecture and elaborate more on the differences between knowledge, knowhow, know-what, know-why, know-who, etc [Coscia, M., Neffke, F. M., & Hausmann, R. (2020). Knowledge diffusion in the network of international business travel. Nature Human Behaviour, 4(10), 1011-1020]. These concepts may not be apparent to most readers."*

As mentioned above, we have re-written the introduction completely. It is now more concise, and it now focuses on the main contributions of our paper: a mathematical model that provides a solid methodological and empirical footing for economic complexity metrics.

Regarding the different instances of knowledge, the Reviewer is correct. We have added a new paragraph (and with it a few new citations) which we hope expands and clarifies the relation of these concepts with our model. Thus, the fourth paragraph in Section 2 ("Model of urban economic complexity") now clarifies this connection after the general approach of the model has been introduced. This new paragraph also addresses comment #4 from the Reviewer, so we provide a more complete answer below.

4. *"On page 2, the authors posited the existence of three underlying variables of economic complexity. I am wondering how these concepts (the intrinsic "complexity" of the industry, a measure of "individual knowhow", and a measure of "collective knowhow") are related to previous literatures. It seems that "individual knowhow" is related to occupation-specific and industry-specific knowledge, and "collective knowhow" is related to location-specific knowledge [see, for example, Jara-Figueroa, C., Jun, B., Glaeser, E. L., & Hidalgo, C. A. (2018). The role of industry-specific, occupation-specific, and location-specific knowledge in the growth and survival of new firms. Proceedings of the National Academy of Sciences, 115(50), 12646-12653]. It may be helpful to clarify these differences as well as commonalities."*

This is a great question for which we do not have a definite answer. Still, the parameter "s_i", which we have labeled "individual knowhow", is necessarily related to skills and

knowhow that are specific to an occupation and/or an industry, in the same sense that Jara-Figueroa et al. use their concepts of occupation-specific and industry-specific knowledge. The other type of knowledge, the location-specific knowledge that Jara-Figueroa consider, is more difficult to connect to our work, because in our model “ r_c ” is a form of “distributed knowhow” that any individual has access to. Ultimately, however, all knowhow is embedded in people, so location-specific knowhow can be internalized by individuals. We have expanded this discussion in a single paragraph that addresses comment #3 and #4 together. It reads as follows:

“Since the model is built upon the concept of a ‘capability’, it is worth briefly discussing what capabilities stand for. The idea is that capabilities are inputs of production, but getting the right capabilities may require different instances of knowledge. Hence, a capability can be acquired either by know-what (e.g., knowing which specific component is needed for a machine to operate), know-why (knowing the causal mechanisms that are critical for the machine to work), know-who (knowing who has a specific expertise to fill in a specific technique), and so on. In general, however, there are reasons to believe that the most determinant component of knowledge is tacit [49, 50, 51]. This type of knowledge is generally referred to as ‘knowhow’. Knowhow can be possessed by the individual in question, or distributed in the city and accessed by living in it and interacting with its infrastructure and its citizens [51]. This is the reason we use ‘knowhow’ as part of the description of the variables derived from the model. In our model, s_i may be associated with occupation-specific and industry-specific knowhow carried by single individuals, while r_c may be associated with distributed knowhow, specific to a location, collectively possessed by the city (see, however, [52] for a study in which each of these types of knowledge is claimed to be internalized by workers and shown to affect patterns of economic diversification; for a review of the factors affecting diversification see [53]).”

5. *“On page 2, Line 83, the authors presented that “The complexity M_f of an industry, however, should not necessarily be associated with the average knowhow of the workers employed in f .” The authors also presented an example. Yet, it remains unclear to me for getting the point [sic]. It would be appreciated if the authors are willing to help further elaborate. The same applies to page 4 Line 89, “Although s_i can be interpreted as the level of schooling or education, it is meant to capture the breadth rather than the depth of knowledge.””*

We thank the reviewer the request for clarification. To address this request, we have added a new figure and we have included additional sentences that should clarify what we mean. The new figure is the following:

The new paragraph explaining M_f reads as follows:

“The parameter M_f represents the “inherent complexity” of the economic activity associated with the production of industry’s product f . The more capabilities are needed, the larger the value of M_f , and the more complex the activity. We emphasize that the model is really agnostic about what capabilities are. For example, the complexity M_f of an industry may not necessarily be associated with the number of employees required by a typical firm, nor with the level of schooling of the workers employed in f (e.g., industries that tend to employ people with PhDs versus industries that employ low skilled workers). Thus, M_f may ultimately quantify something intangible and hard to measure directly in the real world. For example, industries that use low human capital but that are intensive in physical capital may have high M_f if they require the coordination a large set of inputs and materials to be transformed with the use of sophisticated machinery in a complex production process.”

The new paragraph explaining s_i read as follows:

“The parameter s_i represents how many capabilities the individual S_i can acquire on her own. Specifically, it is the probability that she can get any of the M_f capabilities required by the typical business in industry f . This probability can be interpreted as a measure of her individual knowhow. Since capabilities are supposed to vary qualitatively from one another, s_i is meant to capture the breadth rather than the depth of knowledge. It is about how many different things she could know how to do by herself. The explicit inclusion of this parameter in the model represents an important contribution to the framework of economic complexity (cf. [32]). Having said this, since capabilities represent units of knowledge that are interconnected, depth and breadth of knowledge (as measured by her level of schooling, or by the different things she knows how to do, respectively) may be both positively correlated with s_i .”

6. “For all figures, the font sizes of both x-ticks and y-ticks are too small to read, for example, in Figure 1 and Figure 2. The authors are suggested to largely increase the font size of all ticks. In addition, it may be better to label the two panels of Figure 1 with (A) and (B) rather than referencing them using left and right.”

All figures in the manuscript have been revised and have been edited to make them more readable.

7. “In Figure 2, it may be better to add the correlation value and R^2 to the main panels (B) and (C). The same applies to Figure 3 and Figure 4. In the inset of panel (C), there seems to be an outlier around 2001. Could

the authors explain why? In Figure 3, the variable in the title of y-axis starts with (-1). I am wondering if there is a better way to present this or it should go in this way.”

We thank the reviewer for all these observations.

The outlier in 2001 is due to a drop in the employment data in that year. In particular, it seems that a few industries in that year reported very low values of employment, which affected the diversity of cities. That same drop seen in the figure is seen in the time series of average diversity (averaged across cities, by year) [not shown in the manuscript]. The average diversity across cities in most years is around 28, but in 2001 average diversity is 23. Further exploration would be required to see if this was something that actually happened in the US economy (e.g., related to the 2001 dot-com bubble, in which there are reports that state that high-tech industries saw drops in employment after March 2001), and understand which cities and which industries were affected. This is, for the moment, out of the scope of our paper.

Since the panels in these figures have inset figures, adding annotations to the plots risks making them confusing. Thus, we have addressed the Reviewer’s suggestion by adding the correlation value and the R2 in the caption of the figure, rather than in the plots.

Regarding the “(-1)” in the y-axis, we thank the Reviewer for the request for clarification. We have now changed it to just a negative sign, “-“ in the y-axis of the plot, and we have added an explanation in the caption of the figure for why we add a negative sign that says the following:

“We note that we show the negative sign next to Δ_f so that higher levels are interpreted as higher complexities.”

Since we believe this is a potential source of confusion, we have also edited some of the text (see, e.g., new Equation (4)), and we have discussed this negative sign in the main text as follows (see last paragraph in Section 4.4, “Linking the drivers with urban economic performance”):

“[...] we note that we use as independent variables in Eq. (4) the estimates of industry complexity and collective knowhow, Δ_f and γ_c , coming from the results of Section 4.2. Moreover, we note the use of a negative for $\Delta_{f,t}$ and a positive one for $\gamma_{c,t}$ such that their interpretation is consistent with measures of complexity and knowhow.”

8. *“In Table 1 and Table 2, the values of R2 appear to be relatively small, ranging from 0.001 to 0.140. Considering that the regression coefficients of independent variables hold very significant, I am wondering if this is because of the relatively large sample size. Could the authors help understand why R2 is that small, and what this tells us?”*

This is a good question. As the Reviewer rightly notes, large sample sizes can tighten the regression coefficients (i.e., make the standard error of their estimates smaller), without necessarily increasing the R2. In this case, it is worth noting that the reason the R2’s are so small is precisely due to the large sample size we have in those regressions, and the reason we have such a large sample size is because we are regressing observations at the level of city-industry combinations. This is a very disaggregated, high-resolution, dataset. As an

illustration of this, in 2016, across all 19,490 city-industry observations we have, the median number of employees per observation was 772 and the median number of establishments was 51. Predicting the average wage of 772 individuals and the average size of 51 establishments at that level of detail, for 19,490 observations, using only six independent variables (four that are city-specific and two that are industry-specific) is a very hard task, statistically speaking. Hence, it is not surprising that there is a lot of variability still left to be explained. To expand on this observation, we added the following text in Section 5.4 (“Relationship between the drivers and the establishment sizes and wages”):

“Next, we investigate the associations at a fine-grained level of aggregation in which each observation is a city-industry combination, as proposed in Section 4.4 in Eq. (4). At this level of disaggregation, there were 19,490 different city-industry combinations in our data in 2016 (out of $381 \times 78 = 29,718$ possible combinations). The median number of employees per observation was 772 and the median number of establishments was 51. The question is whether our complexity variables can account for some of the variation in the average wages of workers and the average size of establishments across all these observations, and whether they remain statistically significant after adding other competing controls.

Tables 1 and 2 present the results of seven regression models applied to the year $t=2016$, where the dependent variables are logarithm of establishment sizes and wages, respectively. The key observation to note in these tables is that the increase in the level of disaggregation makes the problem of predicting wages and establishment sizes much harder (reflected in the low R^2 values), and yet, industry complexity and the city knowhow remain statistically significant ($p < 0.05$) in the presence of other controls like population size, PCI and ECI.”

9. “In the reference section, the authors are suggested to go through all references and make corrections. For example, to use the same name for PNAS, to add page number for papers published in Scientific Reports and PLOS ONE (and Ref [7] and Ref [20]), to capitalize journal names, to capitalize country names like the US in Ref [8], to lowercase paper titles, and to include the full page ranges (Ref [38]).”

We thank the Reviewer for the attention to detail. We have fixed the references following the suggestions. We expect the Journal to let us know of any other formatting corrections needed.

10. “One the of authors is affiliated with a private firm. Under the conflict of interest, the authors presented “I/We declare we have no competing interests”. It may be necessary to make sure the statement is suitable.”

We appreciate the Reviewer for asking further clarification on this issue. The statement regarding the conflict of interests is accurate as is it. The firm we work in is not involved in this work, nor is involved in this research field. The research we are presenting in this paper started before we joined the firm, and we were postdoctoral researchers. Hence, to confirm, we declare we have no competing interests.

11. “Very minor issues:
(a) Page 2, Line 23, a first-principles model -> a first-principal model.
(b) Page 3, Line 75, there may be a period after “however”.”

We appreciate both edits. We have decided to keep (a) as “first-principles model”, as that is the correct usage of the expression (e.g., compare the results of the following two searches in Google Scholar:

<https://scholar.google.com/scholar?hl=en&q=%22first+principles+model%22> and <https://scholar.google.com/scholar?hl=en&q=%22first+principal+model%22>). Error (b) has been fixed. Thank you.

Reviewer # 2:

We want to start thanking the Reviewer for the initial comment about our paper being “well-written and convinces with comprehensive and elaborated empirical modelling”, and addressing some of the general critiques in the first paragraph of the review.

We appreciate the general comment about the paper being strongly embedded in the literature on urban scaling and cities as complex systems, and not well connected with the broader literature on regional science, economic geography and urban economics. However, some of the perceived “disconnectedness” of our paper with some of the literatures mentioned by the reviewer is deliberate, not because of lack of awareness of others’ work, but mainly due to differences in focus and assumptions across the fields. In our responses below this will become more apparent, and we have made an effort at re-writing many parts of the manuscript to make this point clear. Still, we take the opportunity to address here the following comment from Reviewer #2:

“I am not sure how well it fits the interest of readers of the Royal Society Open Science”

There are several works that have been published in the interdisciplinary journals of the Royal Society (Royal Society Open Science and Journal of the Royal Society Interface) which are connected to our work, and have revealed a broad readership for the themes we discuss in our paper. Papers published in these journals that we cite in our work include the following:

- C. D. Brummitt, A. Gomez-Lievano, R. Hausmann, M. H. Bonds, *Machine-learned patterns suggest that diversification drives economic development*, Journal of the Royal Society Interface 17 (2020) 20190283.
- H. Youn, L. M. A. Bettencourt, J. Lobo, D. Strumsky, H. Samaniego, G. B. West, *Scaling and universality in urban economic diversification*, Journal of The Royal Society Interface 13 (2016).
- Stirling, *A general framework for analysing diversity in science, technology and society*, Journal of the Royal Society Interface 4 (2007) 707–719.
- van Dam, *Diversity and its decomposition into variety, balance and disparity*, Royal Society Open Science 6 (2019) 190452.
- S. Sarkar, E. Arcaute, E. Hatna, T. Alizadeh, G. Searle, M. Batty, *Evidence for localization and urbanization economies in urban scaling*, Royal Society Open Science 7 (2020) 191638.

These papers are not within the mainstream of urban economics and regional science, but touch on issues of how to measure knowhow and/or diversity, and what are the drivers behind differences in economic performance across locations. We believe this list (and the recent publication date of many of the papers) offers evidence that our work will provide new results and insights to many interested researchers sitting at the intersection between economic complexity, urban science, and complex systems.

We now address the specific comments made by Reviewer # 2.

1. There are several critiques in Reviewer’s first comment that merit a separate response for each:
 - a. *“My major concern relates to the overall message of the paper. From the logic of the urban scaling literature, I do understand the purpose of the paper. However, from the perspectives of scholars*

interested in urban development and spatial processes, it is not clear what the paper really offers”.

Indeed, our work originates in the field of economic complexity, and therefore touches on several topics which are the focus of attention in the fields of urban scaling and complex systems, and unsurprisingly it also borders with other fields such as related diversification, economic geography and urban economics. We believe all these fields, while interested in the same types of phenomena, are different enough that connecting them all in our paper is a tough request. Even reviews of the literature such as Hidalgo’s (2021) ("Economic complexity theory and applications") within the field of economic complexity, or Gao et al.’s (2019) ("Computational socioeconomics") within the field of computational and physical sciences applied to social systems, or Content & Frenken’s (2016) ("Related variety and economic development: a literature review") within the field of related diversification, fail to cite and connect all the relevant literatures. We believe the unrealistic request of the Reviewer to make connections to all these fields is due to *our* failure to convey the main goal of our manuscript in a clear way (as the Reviewer suggests when saying that the major concern is with the overall message of the paper). Thus, we have made an extra effort at clarifying what our paper is about, and we have also made our best effort at citing the relevant literature suggested by the Reviewer, discussing the possible connections with other fields.

Due to this comment (and comments from Reviewer #1 that are aligned with this critique), we have completely re-written the Introduction. The most important change is that we have shortened it. We will not quote the full new Introduction in this response, but we do want to highlight some parts of it, which clarify what our paper is about, and why its scope is limited enough that it should not require us to create all possible connections with works in the urban economics and regional science literatures. The introductory paragraphs read as follows (we have put in bold the parts that believe best address the reviewer’s request for clarity about the purpose of our paper):

“Fine-grained representations of the economy of countries, regions and cities in terms of what they produce, and the industries they have, have revealed that the location of economic activities across places is not random [1-6]. Instead, activities tend to locate in cities of different sizes depending on the number of inputs required [7, 8] and co-locate with other activities that require a similar set of inputs [9-14]. **While there is an extensive research program in the traditional economic growth and development literature for understanding how specific inputs, such as natural resources, public goods, labor, physical and human capital, institutions, or amenities, determine the choices of workers and firms for where to locate and what to produce [15], there is less research on more agnostic models that do not specify *a priori* what those inputs are.** The latter has been a big part of the research program in the interdisciplinary field of economic complexity [6], and several dimensionality reduction algorithms have been applied to data to extract ‘metrics of complexity’, presuming to quantify the availability and sophistication of inputs present in an economy [2, 16-22, 6].

[...]

So far, however, these dimensionality reduction algorithms have been hard to connect with theoretical models of how economies work (see, however, [31, 32]). That is, there is no proof that the manipulations of the data that these algorithms conduct actually quantify what they claim to quantify:

the number of capabilities available in an economy, or required by an economic activity. For this reason, their use and interpretation has remained highly contested [33-35]. **Ensuring that the mathematical foundations of economic complexity metrics are robust, that their method of estimation is reliable, and that they stand as meaningful economic indicators is key to understanding cities as complex systems, and acting upon them as such. In this work, we present a novel first-principles model that aims to address these shortcomings. That is, we present how a simple mathematical model can characterize the way inputs combine in cities to generate output (without making strong assumptions about what these inputs may be), how complexity variables then emerge and a method to estimate them, and how they correlate with measures of urban economic performance.”**

We hope the new Introduction sets the proper tone about what to expect from the paper. Still, it is worth repeating some of the message there, which has been also been emphasized by Hidalgo (2021, *Nature Reviews Physics*) and Bustos & Yildirim (2020, *Research Policy*): that traditional models in economics assume a priori what the inputs of production are. Once those inputs are stated, there is a process of data collection to measure proxies for those inputs, and then empirical strategies to quantify (causally, in the ideal case) whether the increase in those inputs affect urban outputs of interest. Hence, urbanization externalities, localization externalities, dynamic externalities, average years of schooling, self-sorting, public goods, information flow, in addition to the classical ones cited in our new Introduction, are just a few among many of the (overlapping) measures/processes that are posited to drive urban output (and urban growth). Even when models assume several unspecified inputs (such as those that use Dixit-Stiglitz production functions), there are several assumptions that are introduced with little support. As mentioned by Krugman (2009, *The Increasing Returns Revolution in Trade and Geography*), “There is no good reason to believe that the assumptions of the Dixit-Stiglitz model – a continuum of goods that enter symmetrically into demand, with the same cost functions, and with the elasticity of substitution between any two goods both constant and the same for any pair you choose – are remotely true in reality.” In our work (like in the work of others in the field of economic complexity) we want to remain more agnostic, not only about the inputs that go into production, but also about how individuals interact with the city. Thus, we have deliberately removed markets and prices. In light of this, the simplicity of our static model is not a bug but a feature. It is also why we believe the model may have applications beyond economics, such as the study of crime and disease (see Patterson-Lomba & Gomez-Lievano, 2018, *arXiv preprint arXiv:1809.00277*, where we present some preliminary analyses applying this model to the case of infectious diseases), and is why Royal Society Open Science is a perfect venue for this paper.

Thus, we do believe our work has contributions that can be valuable for other scholars interested in urban development and spatial processes. If our work is valid (i.e., if it stands our empirical tests that we present in our paper, as well as the tests that other researchers may investigate) scholars interested in urban development and spatial processes may be able to use metrics, such as our measure of collective knowhow, supported by theory, that can be estimated from data, to quantify the complex ability that cities have for turning inputs into outputs, and use it to move forward our understanding of cities. The empirical component of our paper suggests

that what we propose has potential to fulfill this promise.

- b. *“Most importantly, the proposed model does not explain the growth of cities nor their change or development over time.”*

This is a fair and important critique, but we never claimed our model explains growth or change. However, we thank the reviewer for forcing us to be more clear about what the paper does and does not explain.

Our paper used the word “growth” only twice in the main text (in the original version we submitted), and both times the word was used just to say that economic complexity metrics (such as ECI or Fitness) had been shown (by others) to correlate well with measures of economic *growth* (e.g., GDP). We avoided making the claim that our own model explains growth, and when writing the paper, we made the deliberate decision to exclude regressions that attempt to predict/explain growth for that reason. Nevertheless, we concede to the Reviewer that, ultimately, understanding cross-sections of cities is less useful than understanding how cities change in time. Indeed, we agree that part of the purpose of studying cities is to understand their economic growth. We mentioned this limitation explicitly when we had said (in the version that we submitted and which the reviewer read, page 3, line 75) that “[w]e do not, model [...] the dynamics of M, s, or r.” In the new revision, this statement still applies. We have not introduced time in our model. To be more assertive about this, we have added a whole paragraph in the Discussion section expanding on this limitation:

*“Our results have two main limitations which are worth noting, one theoretical and one empirical. First, our model does not yet specify how places acquire capabilities, or how capabilities emerge or evolve. Hence, we currently lack a clear specification for how our model predicts economic change. **That said, preliminary analysis suggests that our complexity variables promise to explain economic growth in cities (see Appendix M for an analysis of how our estimate of collective knowhow correlate with GDP per capita in its level and its growth).** Incorporating time to our model is part of future work [...] The generalization of the model must [...] require an explicit inclusion of technological similarities between industries [97, 98, 12], and presumably include migration dynamics between cities to model the flow of capabilities [99].”*

In the quote above, we have bolded the part in which we say that we now have included empirical analysis of growth in a new Appendix M (“Regressions of GDP per capita, levels and growth”), to draw attention to the Reviewer for some preliminary analysis we have conducted motivated by this comment. Thus, we have explored a few regression specifications to assess the correlation between our metrics and a measure of growth, like the growth rate of GDP per capita.

We start the new Appendix M with a set of regressions that predict the GDP per capita level, and then move to analyze the growth rate of GDP per capita over a 5-year period (from 2010 to 2015), using independent variables at the baseline year. We show the tables of results here, but we refer the Reviewer to the Appendix for

further detail:

Table 5: Cross-sectional linear regressions for (log) GDP per capita at the city level in 2010 as a function of city (log) population size, (log) of share of workers with a graduate degree, industrial diversity of the city, ECI, and the collective knowhow of city.

	Dependent variable:					
	'log(GDP per capita)'					
	(1)	(2)	(3)	(4)	(5)	(6)
'log(Population Size)'	0.099*** t = 8.030					0.053*** t = 4.648
'log(share w/ Grad. degree)'		0.347*** t = 10.623				0.111*** t = 3.355
'Industrial Diversity'			0.011*** t = 4.746			-0.006*** t = -3.115
E.C.I.				0.097*** t = 7.313		0.074*** t = 6.070
'Collective Knowhow'					3.293*** t = 12.301	3.247*** t = 12.617
Constant	9.336*** t = 59.500	11.444*** t = 141.157	10.278*** t = 152.409	10.594*** t = 802.910	10.591*** t = 896.306	10.374*** t = 64.280
Observations	346	345	346	346	346	345
R ²	0.158	0.248	0.061	0.135	0.305	0.533
Adjusted R ²	0.155	0.245	0.059	0.132	0.303	0.526

Note: *p<0.05; **p<0.01; ***p<0.005

As the table above shows, our metric of collective knowhow is the measure that explains the largest amount of the variation in (log) GDP per capita across metropolitan areas in the US in 2016 (similar results apply for all years), with an adjusted R2 of 0.30, and remains statistically significant ($p<0.005$) even in the presence of other variables.

Next, we look at the rate of growth. As is true for most rates of growth, GDP growth is a variable that is difficult to predict. Hence, it is not surprising to observe low R2s. In spite of this, our measure of collective knowhow remains statistically significant ($p<0.005$). To compare the magnitude of the effects of the different covariates, we have standardized the variables to have zero mean and unit variance. Thus, in the table below, the regression coefficients can be compared with one another:

Table 6: Linear regressions for the growth rate of GDP per capita from 2010 to 2015 at the city level as a function of the baseline level of (log) GDP per capita, city (log) population size, (log) of share of workers with a graduate degree, industrial diversity of the city, ECI, and the collective knowhow of city. All independent variables are taken in 2010 and have been standardized to have mean zero and standard deviation equal to one.

	Dependent variable:								
	'GDPpc Growth (5yrs)'								
	(1)	(2)	(3)	(4)	(5)	(6)	(7)	(8)	
'log(GDP per capita)'	0.082 t = 1.531	0.023 t = 0.392	0.108 t = 1.743	0.051 t = 0.927	0.132* t = 2.301	-0.061 t = -0.968	-0.066 t = -0.870	-0.126 t = -1.895	
'log(Population Size)'		0.151** t = 2.596					0.189*** t = 2.852	0.158** t = 2.787	
'log(share w/ Grad. degree)'			-0.052 t = -0.835				-0.063 t = -0.937		
'Industrial Diversity'				0.126* t = 2.286			-0.011 t = -0.164		
E.C.I.					-0.136* t = -2.364		-0.070 t = -1.016		
'Collective Knowhow'						0.260*** t = 4.118	0.242*** t = 3.290	0.265*** t = 4.241	
Constant	-0.000 t = -0.000	-0.000 t = -0.000	-0.000 t = -0.000	-0.000 t = -0.000	-0.000 t = -0.000	-0.000 t = -0.000	0.000 t = 0.000	0.000 t = 0.000	
Observations	345	345	345	345	345	345	345	345	
R ²	0.007	0.026	0.009	0.022	0.023	0.054	0.084	0.075	
Adjusted R ²	0.004	0.020	0.003	0.016	0.017	0.048	0.068	0.067	
Note:							*p<0.05; **p<0.01; ***p<0.005		

Again, our metric of collective knowhow of cities is the covariate that explains the largest amount of variation in the data (adjusted R²=0.048, with the second most explanatory variable being log of population size of cities), and is the variable whose effect magnitude is the largest, even above the reversion to the mean term. As we discuss in Appendix M, the regressions above provide support for interpreting our estimate of collective knowhow as a driver of economic growth.

Since our paper is not about predicting the growth of cities, but rather, it is about providing a solid footing for estimating metrics of complexity, we have left these results in the Appendix, as preliminary results for future work.

c. “It does not establish causal relations (or tests for theses) [sic].”

This is an interesting criticism, but one we believe lies out of the scope of our paper. Causal inference would be required if our paper’s central claims revolved around interventions to change cities. However, we have not discussed any interventions. Our model contributes to model and estimate complexity metrics correctly, and we do this mathematical arguments. The model is a steppingstone toward understanding

the fundamental drivers behind the functioning of cities in a more realistic manner, which should ultimately inform interventions. But we believe at the stage of this model it is still unwise in this paper to avoid establishing causal relationships.

- d. *“From my reading, it presents a number of metrics that correlate with certain features of cities. None of the correlations is surprising or gives insights into aspects that have not been explored in more detailed analyses.”*

We would like to convince the Reviewer to focus less on the correlations and more on the mathematical model. It is because of the model that we claim the metrics we put forward in our paper are different from most of those proposed in the literature (such as ECI and Fitness) that claim to measure economic complexity. Our metrics are not “indices” that one constructs from data using heuristic arguments, or using algorithms from the shelf of machine-learning methods. Instead, our metrics are estimates of fundamental variables that emerge from theory. “Indices” and “estimates” are two very different concepts.

It is true that there are several (perhaps thousands) of indicators that correlate positively with measures of economic performance. Finding just one more indicator is not surprising. The popular economic complexity metrics that have been proposed (ECI and Fitness) belong to this family of “indices” that manipulate data according to some *ad hoc* rules, and end up with a quantity that one can only hope quantifies something meaningful. (A more extreme example is the Human Development Index, which currently is defined as a geometric average of different measures that are assumed to affect human development). In those cases, establishing strong correlations with other metrics becomes the only way of validating the indicators since they lack theoretical support. That is not the case for us. The metrics we present are supported both by theoretical arguments and empirical correlations. The crucial issue in our paper is not that we have constructed a new index to add to the big bag of indices that correlates positively with wages and sizes. To be very precise,

- i. we have a mathematical model from which the metrics emerge from first principles;
- ii. the model reproduces patterns such as urban scaling, the nestedness of activities across places, and the possible sorting of high skill individuals into diverse cities;
- iii. the model itself fits the data better than other competing alternatives;
- iv. the model leads to a strategy for empirically estimating those metrics;
- v. and these metrics end up having all the properties posited by the assumptions of the model (e.g., correlate with population size, diversity, etc.).

Induced by this comment, we have added several new sub-headings into the text, to separate sections, and present with more clarity the contributions listed above. The new revision has the following structure:

Contents	
1	Introduction 4
2	Model of urban economic complexity 4
3	Theoretical and methodological implications of the model 8
3.1	Urban scaling laws 8
3.2	Nestedness of activities across locations 9
3.3	Sorting of individuals with high knowhow 10
3.4	Relationships between the three drivers of urban economic complexity 10
4	Materials and methods 11
4.1	Data 11
4.2	Estimating the drivers of urban complexity from data 11
4.3	Evaluation against competing models 12
4.4	Linking the drivers with urban economic performance 13
5	Results 14
5.1	Comparing prediction power of models 14
5.2	Relation of collective knowhow to population size and industrial diversity 14
5.3	Relation of industry complexity to occupational diversity, and geographical ubiquity 15
5.4	Relationship between the drivers and the establishment sizes and wages 17
6	Discussion and concluding remarks 20
	References 24

e. *“Consequently, I strongly urge the authors to make clear what the paper has to offer that is of interest to general readers outsider their particular literature.”*

With all the revisions mentioned above, we hope we have made clear what the paper has to offer to readers of the Royal Society Open Science.

2. *“In particular in the discussion section, the authors engage in causal interpretations of the relation between their measures and other metrics. For instance, on page 16, line 356, the authors claim that they have shown how “this interaction ... explained WHY a triangular pattern and WHY complex industries and skilled individuals would tend to concentrate in diverse cities”. That is incorrect. The authors do not present a locational choice model for individuals or firms, nor do they explain the emergence of skills or firms. Consequently, they cannot answer the “Why”. Their approach presents a description of the outcome of causal mechanisms without giving insights into these. The authors are advised to reassess their text and stay away from such causal claims or engage in true causal modelling and econometrics that identify the direction of the relations.”*

We agree that the causal language can be problematic and we thank the reviewer for the words of caution. We have modified the language to not give the impression we have measured causal effects. Hence, we have minimized the use of the word “explain” and used instead the word “reproduce”.

The type of patterns we discuss in the Section 3 “Theoretical and methodological implications of the model” have been observed in many urban activities beyond economic ones, and this suggests the underlying mechanisms may be very general and may go beyond markets and optimization choices. Thus, these patterns may also arise from simpler models like ours, and that is part of the point of the paper. From a philosophical point of view, we are invoking different meanings of the words “explanation” and “why”. While distinctions across “levels of explanation” are not common in urban economics and regional science, they are fairly common in sociology and in biology, and they have a long history going back to Aristotle’s “material”, “efficient”, “formal” and “final” causes, and more recently to Niko Tinbergen’s “ontogenetic”, “mechanistic”, “phylogenetic” and “functional” causes, Ernst Mayr’s “proximate” versus “ultimate” causes, and the more complex framework of causation proposed by John Gerring in sociology. These are philosophical notions, but they matter. For example, a lot of research in mathematics, logic, physics and computer science deals with many “why” questions and explanations that have nothing to do with the notions of causality used in the empirical work in economics (which would be classified as “proximate” causes in Mayr’s classification). We expect the diverse readers of the Royal Society Open Science to be comfortable and familiar with all these different levels of explanation. Therefore, we would like to respectfully push back on the presumption of Reviewer #2 that the answers to our “why” questions (e.g., “why is the matrix of cities versus industries triangular?”) *necessarily* come from location choice models, and that answering why questions *necessarily* requires estimating causal effects. To illustrate “why” a triangular pattern emerges from the model, we have added a graphical representation in the new Section 3.2 “Nestedness of activities across locations”:

This is a simulation of our model, and the triangular pattern does not arise from choices by agents or an optimization process, but from the structure of the production function. We hope this is a convincing argument. We have complemented the text already in that part of the manuscript with some additional clarifications:

“The nested (triangular) pattern of industries across cities (see, e.g., [37]) emerges naturally from the function

in Eq. (1). To show that, assume for simplicity that s_i is approximately constant for all individuals i in all locations. Next, let there be two industries L and H such that $M_L < M_H$, and two cities L and H such that $r_L < r_H$. According to Eq. (1), the probability of employment is higher in the city with high collective knowhow in both industries, since $p(M_H, s, r_H) > p(M_H, s, r_L)$ and $p(M_L, s, r_H) > p(M_L, s, r_L)$. In addition, the share of employment in high complexity industries with respect to low complexity industries in diverse cities will be larger than the same ratio in less diverse cities, since $p(M_H, s, r_H)/p(M_L, s, r_H) > p(M_H, s, r_L)/p(M_L, s, r_L)$. The latter property is called “log-supermodularity”, and reflects why diverse locations are disproportionately more competitive in complex economic activities than less diverse cities [66, 46, 19, 8, 67]. This result is shown graphically in Figure 2, where cities and industries have been given different values of r and M , and $p(M, s, r)$ (for constant s) is computed. The triangular pattern that emerges shows that probabilities of employment are nested, as empirically observed [37]. According to the model, it is clear that this nested pattern can become more drastic if we assume individuals with high individual knowhow (high s) tend to locate themselves in cities with high r . [...]

Then, in the section afterwards, we present arguments for extending our model to support (with the addition of some new assumptions) the sorting of individuals with high knowhow into cities of large collective knowhow. We do not reproduce the text here since it was there already in the original version.

3. *“To offer some connections to the well-established literatures on urban growth, the authors are advised to make references to the contemporary literature on “location choice theories”, “urban growth”, and “regional diversification”, all of which have a lot to offer for enriching the paper. In particular the works of location choice and agglomeration economies will provide clearer insights into what locations attract and creates what kind of talent and vice versa (Glaeser, Kallal, Scheinkam, & Shleifer, 1992; Henderson, Kuncoro, & Turner, 1995; Isard, 1956). Similarly, the research strand on related diversification gives detailed insights into how different forms of diversity shape economic structures in a path dependent manner (Boschma, 2017; Boschma & Frenken, 2011). In addition, I recommend the authors to evaluate their work against the traditional ideas of “spatial sorting” that explain processes related to the matching of industries and (skilled) individuals (Mion & Naticchioni, 2009). Given all these works, it is hard to accept that the authors write on page 6: “Presently, we lack a detailed theory about the dynamical laws of these drivers and how they relate to one another””*

We thank the Reviewer for the references. We have *not* added the first references suggested because these classic papers in urban economics (Glaeser et al. 1992; Henderson et al. 1995) deal with dynamics and growth in cities, which, as we have established, are aspects that we have not included in our model or in our analysis. We believe the Reviewer will agree with us for not including the citations if they risk confusing the reader into thinking we claim something we do not show. (We also do not cite Isard 1956 because we were unable to find that reference and read it). On the other hand, we have included the papers by Boschma in the Introduction, since their framework is much more closely related to our framework based on capabilities. The paper by Mion & Naticchioni (2009) is one we did not know before hand, and we thank the Reviewer for pointing it out to us. It is indeed relevant for us and we have included it in Section 3.3.

Finally, we must clarify that our statement *“Presently, we lack a detailed theory about the dynamical laws of these drivers and how they relate to one another”* was meant to refer specifically to the variables in our model, not to the general conceptions of knowledge or knowhow that have been developed in the literature mentioned by the Reviewer. Hence, we have edited the text as follows:

“Presently, we lack a detailed theory about the dynamical laws of **the variables M_f , s_i and rc within our modeling framework**, and how they relate to one another.”

Again, this last statement is a way for us to be transparent about what we can and we cannot claim. We believe our measures and the method we propose to estimate them are useful, but we do not yet feel confident discussing in the main text how cities change in time (even though we know there is an extensive literature on how cities change).

4. *“The authors postulate that someone uses population size “naively” as “proxy for urban knowledge”. Outside the literature on urban scaling, no one is doing that, as Economic Geographers and Regional Scientists are very much aware of the heterogeneity of urban areas across space and time. I suggest the authors to pay attention to this and link their work to these well-established ideas.”*

We think the reviewer is taking the word “naively” out of context. We do not postulate (and therefore we do not cite) that people have used population size as proxy for urban knowledge. The sentence is precisely alerting readers to be careful about the potential uses of population size as a proxy for urban knowhow. We have added some language to remove potential confusion:

“We have shown that our estimates of the knowhow of cities correlate with intuitive measures of urban complexity like population size and industrial diversity, but they also strongly correlate with measures of urban output, like average wages and firm sizes. Crucially, however, our estimates resulted in rankings of cities by collective knowhow that were somewhat surprising, in the sense that large cities like New York or San Francisco MSAs did not rank particularly high. One explanation is that our estimates of urban collective knowhow have a large standard error due to small sample sizes. But another interpretation is that measures like population size (or industrial diversity) alone do not take into account the specific combinations of people and technology in a city that can increase (or decrease) the potential for capability recombination. **This is consistent with the findings by [19], for example, who find that high-complexity patenting tends to be geographically concentrated, yet not exclusively in the most populous cities, or by [103], who find that to understand urban economic performance the specific intra-city composition of economic and social activity and physical infrastructure should be taken into account, in addition to just population size. This suggests that our estimate of city collective knowhow may capture such intra-city composition.”**

The citations 19 and 103 that we cite are:

[19] P.-A. Balland, D. Rigby, *The geography of complex knowledge*, *Economic Geography* 93 (2017) 1-23.
[103] S. Sarkar, E. Arcaute, E. 660 Hatna, T. Alizadeh, G. Searle, M. Batty, *Evidence for localization and urbanization economies in urban scaling*, *Royal Society Open Science* 7 (2020) 191638.

5. *“On page 16, line 340-353, the authors present what the paper contributes. However, they do not discuss what are the benefits of these contributions, i.e., in what sense are they better or more advantageous than what has been previously known?”*

We appreciate the opportunity to clarify the contributions of our work relative to what is previously known. We agree making clear the incremental contributions of each model is crucial for advancing our understanding of cities, and in our paper we have performed careful comparisons to the models that, conceptually and methodologically, are closest to ours. We have added two paragraphs at the beginning of the Discussion that read as follows:

“Work in economic complexity and evolutionary economics has proposed that economic development is the process of accumulating an ever increasing variety of capabilities [93, 2, 12, 56, 6], as opposed to a process of increasing the intensity of a few factors of production. However, since productive capabilities are often embedded in people as tacit knowledge [49, 50, 94, 51], and are therefore not always observable or identifiable, Hidalgo and Hausmann [2] first introduced a set of indicators to infer the number of capabilities present in a place (and required by firms in different economic activities) by analyzing directly the high-dimensional data about the collection of products that places are able to produce. They named these indicators the Economic Complexity Index (ECI) and the Product Complexity Index (PCI). Soon after, additional research followed suit and developed other algorithms to use such high-dimensional data to compute complexity metrics (e.g., [16, 22, 20]).

While the idea of economic performance as a matter of capability accumulation is well rooted in theories of cultural and social evolution [62], and is consistent with the observation that differences in economic performance across time and space seem to arise because some societies are collectively more productive than others [95, 62, 96], the economic complexity indices proposed to this date rest on less solid ground, and their use remains therefore contested [33, 97, 34]. So far, there is no theory that demonstrates that the mathematical manipulations that underlie these algorithms actually quantify anything like ‘number of capabilities’. Schetter [31] has shown that ECI (or a measure closely related to it) may rank places according to some underlying productivity (conditional on some assumptions about the structure of the data), but others have shown that ECI is better interpreted as an index to quantify how much a place is specialized on a few economic activities [98, 21, 99]. If we believe the economic performance of places is related to the number of capabilities present in them, then we must create reliable ways of estimating them. As indices of economic complexity become more commonly used in the policy sphere [100, 101, 102], we must ensure to have sensible theories, robust methodologies, and reliable interpretations of metrics that actually quantify capabilities. In this paper, we have proposed a mathematical model to address these issues.”

6. *“In the end, the authors write that the major conclusion of the paper is that “both the number and the type of capabilities affect economic performance. The generalization of the model must thus require an explicit inclusion of technological similarities between industries ...” (p. 18, line 416). This is very much what the literature on related variety argues for a long time (Boschma & Frenken, 2011; Frenken & Boschma, 2007; Frenken, Van Oort, & Verburg, 2007). I am surprised to see no reference to this literature and discussion on what this implies for the future development of the proposed approach.”*

We thank the Reviewer for this comment. It is indeed the case that we are proposing to further develop our mathematical model to accommodate the observations that have been made in the related variety and related diversification literatures (advanced by Boschma, Frenken, and collaborators). We share the views of these authors and their conceptual frameworks, and we believe mathematical formalisms like the one we presented in our work (a) can contribute to that literature, and (b) should incorporate what that literature has developed conceptually, and what it has observed empirically. Thus, we have added several citations we have read to be relevant, which should convey this message better:

“The generalization of the model must thus require an explicit inclusion of technological similarities between industries as has been studied in the literature of related variety and related diversification [11, 12, 94, 108, 109, 110], and presumably include migration dynamics between cities to model the flow of capabilities (e.g., [111, 112, 113, 52]).”

The citations mentioned above are the following:

[11] R. Boschma, K. Frenken, The emerging empirics of evolutionary economic geography, *Journal of Economic Geography* 11 (2011) 295–307.

- [12] R. Boschma, Relatedness as driver of regional diversification: A research agenda, *Regional Studies* 51 (2017) 351–364.
- [52] C. Jara-Figueroa, B. Jun, E. L. Glaeser, C. A. Hidalgo, The role of industry-specific, occupation-specific, and location-specific knowledge in the growth and survival of new firms, *PNAS* 115 (2018) 12646–12653.
- [94] F. Neffke, M. Henning, Skill relatedness and firm diversification, *Strategic Management Journal* 34 (2013) 297–316.
- [108] R. Boschma, K. Frenken, Technological relatedness, related variety and economic geography, in: *Handbook of regional innovation and growth*, Edward Elgar Publishing, 2011.
- [109] K. Frenken, F. Van Oort, T. Verburg, Related variety, unrelated variety and regional economic growth, *Regional Studies* 41 (2007) 685–697.
- [110] K. Frenken, R. A. Boschma, A theoretical framework for evolutionary economic geography: industrial dynamics and urban growth as a branching process, *Journal of economic geography* 7 (2007) 635–649.
- [111] D. Bahar, R. Hausmann, C. A. Hidalgo, Neighbors and the evolution of the comparative advantage of nations: Evidence of international knowledge diffusion?, *Journal of International Economics* 92 (2014) 111–123.
- [112] F. Neffke, M. Hartog, R. Boschma, M. Henning, Agents of structural change: The role of firms and entrepreneurs in regional diversification, *Economic Geography* 94 (2018) 23–48.
- [113] R. Boschma, V. Martin, A. Minondo, Neighbour regions as the source of new industries, *Papers in Regional Science* 96(2017) 227–245.

7. *“There are some claims in the paper that are hard to accept and that just seem to be made to make [sic] the model fit. For instance, on page 9, line 230, it is claimed that “more years of educational attainment are usually a measure of specialization, but they are also an indication of a person’s competences to perform several tasks”. Besides the fact that both parts of the sentence somewhat contradict each other, I also don’t believe the second part.*

The authors need to add some empirical support or references to support their claim. Similar is true for the claim on page 5, line 127, that diverse cities are “disproportionately more competitive in complex economic activities”. Please add support for this or remove.”

This is a fair criticism, and we agree. In fact, we had already noted the caveats of this assumption in the original text (in the same paragraph mentioned by the reviewer): “Whether years of schooling is a good proxy for individual knowhow, however, remains an empirical question (see, e.g., 39). Bearing in mind these caveats, we will proxy $\beta_{c,t}$ using the average levels of schooling in the city c in year t (see Appendix E for details).” Thus, we emphasize here that this is an empirical question. The original reference “39” cited there (Bacolod et al, 2010) explores (in large part, but among other things) this empirical question, and argues that skills and education are different. We believe the Reviewer and us both agree with this conclusion. However, Bacolod et al. also show that education and skills may correlate positively when they drop certain skills from their analysis (such as motor skills). Thus, education may be a useful proxy for skill under some circumstances. Beyond this issue is the question of why we did not apply our “fixed effects methodology” to estimate the parameter β_i in our model, and why did we have to use a (restricted) proxy. The simple answer is that we do not have individual-level data to do this. Hence, to keep our analysis faithful to the structure of the model, in which there are three quantities affecting the probability of production (an industry-specific, and individual-specific, and a city-specific quantities, multiplied in the exponent of an exponential function), we needed to include a proxy for individual-specific quantity.

We do not want to deviate the attention of the reader into a deep discussion about how to proxy individual skills, however. It should be clear from our model that if individual data is

available, and our model is correct (two big assumptions, granted), the method we propose should estimate skills, and proxies like education would not be needed. But our lack of individual-level data should not undermine the results we present in our paper. We have made the decision to include the best next option, the average years of schooling as a proxy for skills, instead of leaving out any measure of individual knowhow from our analysis due to this limitation in our data (and risk potential omitted variable bias). As the reviewer can attest, our analysis does not emphasize nor discuss to a large extent the meaning and significance of this variable throughout the text. The focus is instead on the collective knowhow of cities and complexity of industries.

Still, we have added extra words of caution to the last part of that paragraph that the reviewer:

“Thus, we emphasize that statements, hypotheses, and propositions about the relation of our model with how this driver [as individual knowhow] affects wages and establishment sizes should be taken with caution.”

Finally, regarding the comment about adding citations to support some of our statements, we now have the following:

“The latter property is called “log-supermodularity”, and reflects why diverse locations are disproportionately more competitive in complex economic activities than less diverse cities [66, 46, 19, 8, 67]. This result is shown graphically in Figure 2, where cities and industries have been given different values of r and M , and $p(M,s,r)$ (for constant s) is computed. The triangular pattern that emerges shows that probabilities of employment are nested, as empirically observed [37]”

The references (we know of) that show that innovation and other sophisticated economic activities disproportionately concentrate in diverse cities are:

- [66] M. P. Feldman, D. B. Audretsch, Innovation in cities: Science-based diversity, specialization and localized competition, *European economic review* 43 (1999) 409–429.635
- [67] D. Bahar, H. Rapoport, R. Turati, Birthplace diversity and economic complexity: Cross-country evidence, *Research Policy* (2020) 103991.
- [8] P.-A. Balland, C. Jara-Figueroa, S. G. Petralia, M. P. Steijn, D. L. Rigby, C. A. Hidalgo, Complex economic activities concentrate in large cities, *Nature Human Behaviour* (2020) 1–7.
- [19] P.-A. Balland, D. Rigby, The geography of complex knowledge, *Economic Geography* 93 (2017) 1–23.
- [37] S. Bustos, C. Gomez, R. Hausmann, C. A. Hidalgo, The Dynamics of Nestedness Predicts the Evolution of Industrial Ecosystems, *PLoS ONE* 7 (2012) e49393.
- [46] D. R. Davis, J. I. Dingel, The comparative advantage of cities, *Journal of International Economics* (2020) 103291.

8. “Moreover, it is recommended to check the rest of the paper and remove or back-up such claims.”

We have reviewed the text thoroughly and added/removed relevant language, and included several new citations to back-up our claims. Many have already been mentioned in our responses above due the previous comments from both Reviewers. Given the long list of places where we have added citations, we only mention here that 53 new citations were included (i.e., the number of citations went from 68 to 121). We do not claim our references are exhaustive, but we hope they point readers to connections to others’ work, and to papers

which can serve as guide for further exploration.

9. *“Please point out that the empirical results are restricted to the US, which is known to be non-representative for most countries in the world.”*

We thank the Reviewer for noting this limitation. It is indeed an important limitation, and we have added it to the limitations paragraph in the Discussion section:

“[...] we must highlight that we have tested the validity of our model using data from US metropolitan areas. While the US urban system has been predominantly the focus of study for understanding cities, there is evidence that the regularities found in it are not necessarily universal (see, e.g., [114, 115]). Hence, applying this model and the method of estimating economic complexity metrics to explain urban outcomes in other parts of the world stands as an important part of future research.”

There are probably several citations that could be included here. However, we provided two which we think are representative, and link to other references in their bibliography. The citations mentioned are:

[114] J. P. Chauvin, E. Glaeser, Y. Ma, K. Tobio, What is different about urbanization in rich and poor countries? Cities in Brazil, China, India and the United States, *Journal of Urban Economics* 98 (2017) 17–49.

[115] A. J. Venables, Breaking into tradables: Urban form and urban function in a developing city, *Journal of Urban Economics* 98 (2017) 88–97

10. *“In the beginning of the work, e.g., page 3, the paper reads very much like “complexity” being more or less the same as “diversity”. This is of course not correct and the authors need to revise this part to avoid this impression.”*

We very much appreciate this comment from Reviewer #2 because it is an interesting discussion, and our model presents a particular stance in this discussion about diversity versus complexity. Thanks to this comment, we have reviewed the manuscript (in particular page 3), and we (still) think the words complexity and diversity are appropriately used. However, due to the potential for confusion, and given the distinction often made between these two concepts, we have included the following new text as a footnote at the end of Section 2, “Model of urban economic complexity”, and we have included other references that either make assertions similar or opposite to ours:

“³ A note about terminology may be necessary here. All three measures increase in value with the number of distinct capabilities considered. Hence, they can be thought of as measures of diversity (or variety, as defined by [57]; see also [58]). The more capabilities are required by an industry, or possessed by an individual or by a city, the more possibilities of recombining capabilities there are (for example, rc itself has the potential to create a constant push toward greater rc as argued in Appendix C). This aspect of the model supports the interpretation of these quantities as measures of complexity as well (in the same vein that [59] characterize how biological diversity and complexity go hand-in-hand), despite the fact that diversity and complexity are typically considered to measure different aspects of a system (for a thorough review of these concepts see [60]).”

The references cited are:

[57] A. Stirling, A general framework for analysing diversity in science, technology and society, *Journal of the*

Royal Society Interface 4 (2007) 707–719.620

[58] A. Van Dam, K. Frenken, Variety, complexity and economic development, Research Policy (2020) 103949.

[59] D. W. McShea, R. N. Brandon, Biology's First Law, University of Chicago Press, 2010.

[60] S. Page, Diversity and Complexity, Princeton University Press, 2011.

11. *“With respect to the empirics, I recommend to at least include tests for spatial autocorrelation and cluster standard errors at the MSA level, to account for the fact that the observations are not independent of each other.”*

We agree with both statements: clustered errors by MSA are appropriate since industries tend to co-locate within them, and spatial autocorrelations are likely to be present across MSAs. We take this opportunity to highlight that the focus of the paper is to present a theoretical framework to estimate economic complexity parameters, and show that they make sense in general, not to answer a particular economic question about the US economy of cities and their particular statistics. Hence, we do not want to deviate the attention of readers into thinking that our paper is about explaining US wages or establishment sizes. As the Reviewer knows, adding a test of spatial autocorrelation is not immediate and there is no standard way of doing it: adding such analysis will require a significant effort in the programming, in defining a distance metric/neighborhood criterion, in choosing one among the many statistical tests (Moran's I, Anselin's I, Ripley's K, Getis' G, etc.), in selecting one of the different regression models (spatial lag models versus spatial error models), etc.. Similarly, adding clustered standard errors are more likely to increase the standard errors of the coefficients more than the sign and size of the coefficients. Given the stringent levels of significance we have used [$p < 0.005$], we do not expect big qualitative changes to our results. Thus, even though we thank the reviewer for pointing out this limitation, respectfully, we would like to push back on this recommendation for analysis that is not central to the contributions of our paper.

Still, in Section 4.4., “Linking the drivers with urban economic performance”, we thought it was appropriate to add the following words of caution due to Reviewer's comment on spatial autocorrelation:

“For simplicity of presentation, we have not included diagnoses of spatial autocorrelations. However, researchers seeking to explain urban outcomes should make use of these tests, since cities are not independent of one another.”

Conclusion

We thank again for the positive comments we received about our manuscript from both reviewers. Above all, we appreciate the detailed comments and suggestions. The manuscript has gained immensely from them, and we hope we have fully addressed all their concerns. With these changes we believe the manuscript is ready to be accepted for publication.